# Genome streamlining in a minute herbivore that manipulates its host plant

Robert Greenhalgh[1†], Wannes Dermauw[2†‡*], Joris J Glas[3§],
Stephane Rombauts[4,5], Nicky Wybouw[2], Jainy Thomas[6], Juan M Alba[3],
Ellen J Pritham[6], Saioa Legarrea[3], René Feyereisen[2,7], Yves Van de Peer[4,5,8],
Thomas Van Leeuwen[2], Richard M Clark[1,9*], Merijn R Kant[3*]

[1]School of Biological Sciences, University of Utah, Salt Lake City, United States;
[2]Laboratory of Agrozoology, Department of Plants and Crops, Faculty of Bioscience
Engineering, Ghent University, Ghent, Belgium; [3]Department of Evolutionary and
Population Biology, Institute for Biodiversity and Ecosystem Dynamics, University of
Amsterdam, Amsterdam, Netherlands; [4]Department of Plant Biotechnology and
Bioinformatics, Ghent University, Ghent, Belgium; [5]Center for Plant Systems
Biology, VIB, Ghent, Belgium; [6]Department of Human Genetics, University of Utah
School of Medicine, Salt Lake City, United States; [7]Department of Plant and
Environmental Sciences, University of Copenhagen, Copenhagen, Denmark; [8]Centre
for Microbial Ecology and Genomics, Department of Biochemistry, Genetics and
Microbiology, University of Pretoria, Pretoria, South Africa; [9]Henry Eyring Center
for Cell and Genome Science, University of Utah, Salt Lake City, United States

**\*For correspondence:**
wannes.dermauw@ugent.be (WD);
clark@biology.utah.edu (RMC);
M.Kant@uva.nl (MRK)

[†]These authors contributed
equally to this work

**Present address:** [‡]Flanders
Research Institute for
Agriculture, Fisheries and Food
(ILVO), Plant Sciences Unit,
Merelbeke, Belgium; [§]Rijk Zwaan
Breeding BV, De Lier, The
Netherlands

**Competing interest:** See
page 32

**Reviewing editor:** Detlef
Weigel, Max Planck Institute for
Developmental Biology,
Germany

**Abstract** The tomato russet mite, *Aculops lycopersici,* is among the smallest animals on earth. It is a worldwide pest on tomato and can potently suppress the host's natural resistance. We sequenced its genome, the first of an eriophyoid, and explored whether there are genomic features associated with the mite's minute size and lifestyle. At only 32.5 Mb, the genome is the smallest yet reported for any arthropod and, reminiscent of microbial eukaryotes, exceptionally streamlined. It has few transposable elements, tiny intergenic regions, and is remarkably intron-poor, as more than 80% of coding genes are intronless. Furthermore, in accordance with ecological specialization theory, this defense-suppressing herbivore has extremely reduced environmental response gene families such as those involved in chemoreception and detoxification. Other losses associate with this species' highly derived body plan. Our findings accelerate the understanding of evolutionary forces underpinning metazoan life at the limits of small physical and genome size.

## Introduction

The free-living microarthropod *Aculops lycopersici* (Tryon) belongs to the superfamily of the Erio-phyoidea (Arthropoda: Chelicerata: Acari: Acariformes) that harbors the smallest plant-eating animals on earth (*Keifer, 1946*; *Navia et al., 2010*; *Sabelis and Bruin, 1996*). Eriophyoids are known by many names including gall, blister, bud, and rust mites, depending on the type of damage they cause (*Hoy, 2004*). Since the 1930s, the tomato russet mite *A. lycopersici* has been reported as a minor pest of cultivated tomato (*Solanum lycopersicum* L.) worldwide (*Massee, 1937*). For unknown reasons, it has emerged in recent years as a significant pest of tomatoes in European greenhouses (*Moerkens et al., 2018*). While it is extremely small – only ~50 μm wide and 175 μm in length (*Figure 1a,b*) – it can reach high population densities (*Figure 1c*). The damage it causes to plants superficially resembles that of microbial disease (*Figure 1d*), for which it is often misdiagnosed, and controlling it is troublesome (*Gerson and Weintraub, 2012*; *Van Leeuwen et al., 2010*).

**eLife digest** Arthropods are a group of invertebrates that include insects – such as flies or beetles – arachnids – like spiders or scorpions – and crustaceans – including shrimp and woodlice. One of the tiniest species of arthropods, measuring less than 0.2 millimeters, is the tomato russet mite *Aculops lycopersici*. This arachnid is among the smallest animals on Earth, even smaller than some single-celled organisms, and only has four legs, unlike other arachnids. It is a major pest on tomato plants, which are toxic to many other animals, and it feeds on the top cell layer of the stems and leaves. Tomato growers need a way to identify and treat tomato russet mite infestations, but this tiny species remains something of a mystery.

One way to tackle this pest may be to take a closer look at its genome, as this could reveal what genes the mite uses to detoxify its diet. Examining the mite's genome could also reveal information about how evolution handles creatures becoming smaller. An area of particular interest is the overall size of its genome. Not all of the DNA in a genome is part of genes that code for proteins; there are also sections of so-called 'non-coding' DNA. These sequences play important roles in controlling how and when cells use their genes. In the human genome, for example, just 1% of the DNA codes for protein. In fact, most human protein-coding genes are interrupted by sequences of non-coding DNA, called introns.

Here, Greenhalgh, Dermauw et al. sequence the entire tomato russet mite genome and reveal that not only is the mite's body size miniature: these tiny animals have the smallest arthropod genome reported to date, almost a hundred times smaller than the human genome. Part of this genetic miniaturization seems to be down to massive loss of non-coding DNA. Around 40% of the mite genome codes for protein, and 80% of its protein coding genes contain no introns. The rest of the miniaturization involves loss of genes themselves. The mites have lost some of the genes that determine body structure, which could explain why they have fewer legs than other arachnids. Additionally, they only carry a small set of genes involved in sensing chemicals and clearing toxins, which could explain why they are mostly found on tomato plants.

Greenhalgh, Dermauw et al.'s findings shed light on what may happen to the genome at the extremes of size evolution. Sequencing the genomes of other mites could reveal when in evolutionary history this genetic miniaturization occurred. Furthermore, a better understanding of the tomato russet mite genome could lead to the development of methods to detect the infestation of plants earlier and be highly beneficial for tomato agriculture.

The mite feeds on plant epidermal cells (*Royalty and Perring, 1988*), which are relatively low in nutrients, with needle-shaped mouth parts (stylets) that allow the transfer of saliva and the uptake of cell contents (*Nuzzaci and Alberti, 1996*). The first visible signs of a russet mite infestation are a rapid local collapse of the leaf hairs (trichomes) on the stem, leaflet or petiole upon which the mites are feeding (*van Houten et al., 2013*). This is followed by withering and necrosis of infested leaves, which ultimately leads to a bronzed or russet color, from which the mite owes its name (*Jeppson et al., 1975*; *Kawai and Haque, 2004*). Although it is now a global pest on tomato, it can survive on many related solanaceous plants (nightshade family) such as potato, tobacco, petunia, nightshade, and various peppers (*Perring and Farrar, 1986*), as well as on a few hosts outside the nightshade family (*Perring and Royalty, 1996*; *Rice and Strong, 1962*).

The Eriophyoidea belong to the Chelicerata, a subphylum of Arthropoda which includes spiders, scorpions, ticks, and mites. The Eriophyoidea consists of three families – Phytoptidae, Eriophyidae (or eriophyids, to which *A. lycopersici* belongs), and Diptilomiopidae, and comprises 357 herbivorous genera found on more than 1800 different plant species (*Oldfield, 1996*; *Zhang, 2011*). Eriophyoids are known to manipulate host plant resource allocation and resistance, and many species do so by inducing the formation of plant galls (*de Lillo et al., 2018*), possibly by secreting molecular mimics of plant hormones in their saliva (*De Lillo and Monfreda, 2004*; *de Lillo and Skoracka, 2010*). Although *A. lycopersici* is not a gall-inducing species, it nevertheless manipulates the defense mechanisms of its tomato host to its benefit. Through an unknown mechanism during feeding, this mite suppresses the jasmonic acid (JA) signaling pathway (*Glas et al., 2014*; *Schimmel et al., 2018*). This blocks the ability of the tomato host plant to produce defensive metabolites and proteins against

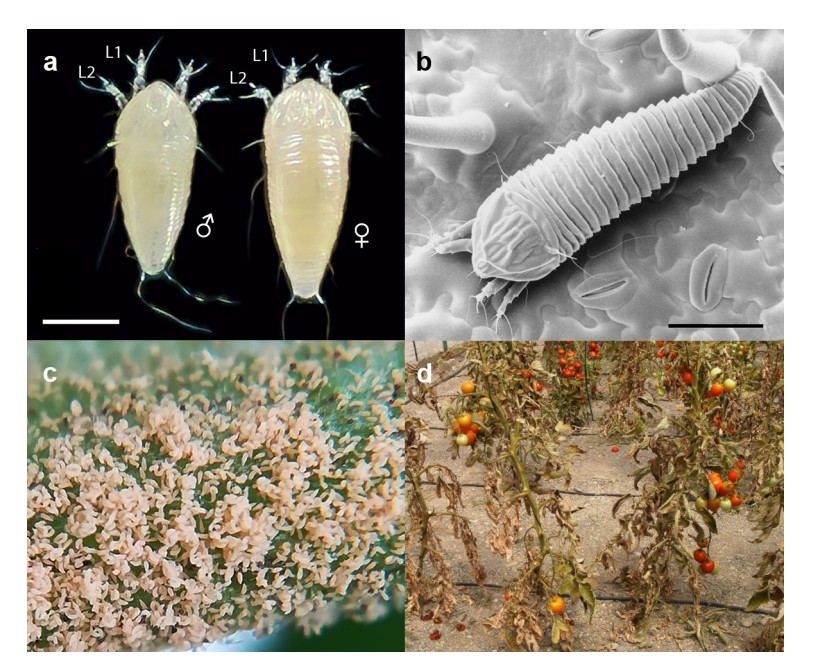

**Figure 1.** The tomato russet mite *Aculops lycopersici* is a devastating pest of tomato. (**a**) Habitus of the eriophyoid mite *A. lycopersici*. Male (left) and female (right) mites are slender, worm-like animals bearing, in contrast to non-eriophyoid mites with four pairs of legs, only two pairs of small legs (indicated by L1 and L2). (**b**) Low temperature (LT) - scanning electron microscopy (SEM) image of *A. lycopersici* on a leaf of *S. lycopersicum*. (**c**) *A. lycopersici* populations can rapidly build to extremely large numbers on tomato stems and leaves. (**d**) *A. lycopersici* damage of heavily infested tomato plants is shown. Scale bars in panels a and b represent 0.05 mm.

herbivorous insects and mites (*Alba et al., 2015*; *Howe and Jander, 2008*), thereby rendering the plant defenseless. The consequences of suppressing host defenses for the herbivore's selective environment may be variable depending on the degree of host specialization (*Blaazer et al., 2018*; *Kant et al., 2015*) but for mite species that can feed on multiple hosts, there are indications of a trade-off between the ability to suppress defenses and the ability to cope with xenobiotics (*Kant et al., 2008*; *Wybouw et al., 2015*). Many species of eriophyoid mites cause little damage to their hosts (*Jeppson et al., 1975*), or alternatively induce damage indirectly as vectors of pathogens (*Navia et al., 2013*). In contrast, while *A. lycopersici* is not known to vector plant diseases, its ability to alter the chemistry and morphology of tomato severely weakens the plants, which are then overwhelmed and killed by exponentially growing *A. lycopersici* populations (*Figure 1c,d*; *Perring, 1996*).

In addition to being a priority pest of tomato, *A. lycopersici* and related eriophyoids are among the most extreme examples of miniaturization in arthropods. As one of the smallest documented animal species (*Bailey and Keifer, 1943*), with dimensions smaller than some single-celled organisms (*Polilov, 2015*), it is not surprising that *A. lycopersici* has a derived morphology. Compared to almost all adult arachnids outside of the Eriophyoidea, which have a body plan with eight legs, *A. lycopersici* has only four legs (*Figure 1a,b*). Further, reproductive structures, which are located at the terminal end in other mites, are positioned in the central ventral region (*Nuzzaci and Alberti, 1996*). This type of morphology has resulted in altered reproductive behavior wherein males, instead of direct insemination, deposit spermatophores (packets of sperm) in the environment that are subsequently picked up by females (*Al-Azzazy and Alhewairini, 2018*; *Oldfield and Michalska, 1996*). Despite these morphological and behavioral innovations, *A. lycopersici* retains the haplodiploid mechanism of sex determination characteristic of many other mite species (*Anderson, 1954*). Further, female *A. lycopersici* mites can lay up to four eggs per day, and the generation time is as little as 5 days under optimal conditions (*Kawai and Haque, 2004*; *Rice and Strong, 1962*). These features, which resemble those of other agriculturally important mite herbivores, result in rapid

overexploitation of the host plant and have undoubtedly contributed to the importance of this species as a field and greenhouse pest of tomato.

Here, we present the genome of *A. lycopersici*, the first for an eriophyoid mite. At only 32.5 Mb, it is the smallest arthropod genome reported to date (*Grbić et al., 2011*; *Waldron et al., 2017*). As revealed by contrasting the genomic architecture of the tomato russet mite with other sequenced arthropods, including the two-spotted spider mite *Tetranychus urticae* (*Grbić et al., 2011*), a generalist herbivore often found in co-infestations alongside *A. lycopersici* (*Glas et al., 2014*), we elucidate mechanisms underlying dramatic genome reduction. In particular, we observed typical features of streamlined genomes (*Arkhipova, 2018*; *Hessen et al., 2010a*), including a marked reduction in the distance between adjacent genes, and few repetitive sequences. Massive loss of introns was apparent. Moreover, reductions in specific genes and gene families, such as environmental response genes, associate with *A. lycopersici*'s ability to suppress host plant defenses as well as its derived morphology. The genome therefore sheds light not only on mechanisms of extreme metazoan genome reduction, but also on the interplay between gene content and the lifestyle of small herbivores that manipulate their environment.

## Results

### Genome size, assembly, and annotation

We assembled the genome of *A. lycopersici* into seven scaffolds of cumulative length 32.53 Mb, of which 99.98% is represented on scaffolds 1–5 of lengths 12.44, 10.50, 3.66, 3.57 and 2.36 Mb, respectively. The remaining two scaffolds are each <6 kb in length, in addition to a mitochondrial genome scaffold. The observed assembly length is similar to the length estimated by a k-mer analysis with genomic sequence reads (34.81 Mb). Separate genome completeness estimates with CEGMA (*Parra et al., 2007*) and BUSCO (*Simão et al., 2015*) located 90.7% and 86.0% of the expected core eukaryotic genes, respectively; these values are within the same range as those for *T. urticae*, the only other sequenced chelicerate herbivore, and for which a high-quality Sanger assembly is available (95.16% and 92.07%, respectively). As an additional assessment of completeness, we generated a de novo assembly of the *A. lycopersici* transcriptome using deep, paired-end Illumina RNA-seq reads derived from mixed sex and developmental stages, and aligned it to the genome sequence. We found that 98.2% of transcript contigs could be located on the reference sequence. Of the remaining 243 unplaced transcript sequences, only eight had similarity to known arthropod sequences; the others had homology to bacterial, fungal, or plant sequences, or lacked homology to sequences in existing databases.

### Features of extreme genome reduction in *A. lycopersici*

Annotation of the *A. lycopersici* genome by automated methods, coupled with extensive manual curation, revealed only 10,263 protein-coding genes. As assessed against other mite genomes, including *T. urticae*, *Dermatophagoides pteronyssinus* (the European house dust mite) (*Waldron et al., 2017*), and *Metaseiulus occidentalis* (a phytoseiid predatory mite) (*Hoy et al., 2016*), as well as the *Drosophila melanogaster* and human genomes, several features of genic organization in *A. lycopersici* stand out (*Table 1*). The fraction of the genome comprising coding sequence is highest in *A. lycopersici*, and the distance between genes is the lowest. Associated with the compact genic landscape of *A. lycopersici* (*Figure 2* and *Figure 2—figure supplements 1–6*), the percentage of the genome consisting of transposable elements was merely 1.54%, which is more than fourfold less than that observed in several other mite genomes, or in the insect *D. melanogaster* (*Figure 2—figure supplement 1*, *Supplementary file 1* — 'Table S1' Tab). Nevertheless, sequences homologous to the major classes of transposable elements, such as DNA transposons, including *Helitrons*, as well as both long terminal repeat (LTR) and non-LTR retrotransposons, were detected (*Supplementary file 1* — 'Table S1' Tab and 'Table S2' Tab). Across the *A. lycopersici* genome, extended regions of low genic composition and high TE density were not observed (*Figure 2—figure supplement 2*), consistent with the purported holocentric chromosome architecture (lack of regional centromeres) of eriophyoid mites (*Helle and Wysoki, 1996*).

We also observed that the *A. lycopersici* genome has only 3057 introns in coding sequences (CDS introns), which is more than an order of magnitude fewer than the 44,881 in the 90 Mb *T. urticae*

**Table 1.** Genome metrics for *A. lycopersici,* other mite species, *D. melanogaster* and *H. sapiens.*

| Species | Genome size (Mb) | PCG* | % intronless† | Coding %‡ | Intergenic %§ | Intronic %¶ | Intergenic M | Intron M |
|---|---|---|---|---|---|---|---|---|
| *A. lycopersici* | 32.53 | 10,263 | 83.67 | 42.26 | 45.12 | 12.62 | 538 bp | 170 bp |
| *D. pteronyssinus* | 70.76 | 12,530 | 25.29 | 35.26 | 46.00 | 18.73 | 542 bp | 75 bp |
| *T. urticae* | 90.83 | 19,086 | 18.26 | 22.10 | 54.12 | 23.78 | 1302 bp | 94 bp |
| *M. occidentalis* | 151.90 | 17,310 | 24.97 | 15.25 | 59.14 | 25.61 | 2035 bp | 135 bp |
| *D. melanogaster* | 143.73 | 13,931 | 16.37 | 15.60 | 57.37 | 27.03 | 1228 bp | 69 bp |
| *H. sapiens* | 3088.27 | 19,636 | 6.74 | 1.10 | 68.14 | 30.77 | 23,279 bp | 1,505 bp |

*PCG: protein coding genes.

†Percent coding genes with no introns.

‡Percentage of genome in coding regions.

§Percentage of genome in between genes.

¶Percentage of genome in introns.

M = Median. See 'Genome metric calculations' in Materials and methods and **Table 1—source data 1** for more information.

The online version of this article includes the following source data for Table 1:

**Source data 1.** GFF3 annotation file of the *A. lycopersici* genome.

genome, and the 35,841 in the 70.8 Mb *D. pteronyssinus* genome. Strikingly, nearly 84% of *A. lycopersici* protein coding genes were intronless, which is more than threefold higher than observed for the other mite species we analyzed, and more than fivefold higher than for *D. melanogaster* (**Table 1**). To further investigate the dynamics of intron evolution, we evaluated patterns of intron gain and loss in orthologous genes among *A. lycopersici* and 17 other animal genomes using the Malin analysis pipeline (**Csurös, 2008**; **Figure 2**, and **Figure 2—figure supplements 3** and **4**, and **Supplementary file 2**). At 29,447 conserved intron sites (**Figure 2a**), *A. lycopersici* has a mere 207 introns. This is an ~11 fold reduction from that seen in the species with the next lowest counts, the European house dust mite *D. pteronyssinus*, at 2292. Apart from *A. lycopersici*, Acari intron loss rates were broadly similar to those observed for other arthropods, except for *M. occidentalis*, for which high rates of both intron loss and gain were apparent, a finding previously reported (**Hoy et al., 2016**). However, the rate of intron loss in *A. lycopersici* was higher than observed in *M. occidentalis* (**Figure 2b**), and in contrast to *M. occidentalis*, intron gains were minimal (**Figure 2—figure supplement 4**). The only evidence for retention of the minor spliceosome in *A. lycopersici* comes from the presence of a single canonical U12 (minor) intron in the gene *aculy03g00270* that encodes an ultra-conserved calcium channel (splice sites AT-AC in intron one of length 12.5 kb). Splicing of this large intron is supported by RNA-seq read alignments, and the orthologous intron one of the *T. urticae* orthologue of this gene is one of the three U12 introns documented previously in *T. urticae* (**Grbić et al., 2011**).

Although relatively few conserved introns are present in the *A. lycopersici* genome, they exhibit a bias toward 5' gene ends (**Figure 2—figure supplement 5**), and compared to most arthropods, the median intron length is larger (**Table 1** and **Figure 2—figure supplement 6**). In a single copy (orthologous) gene set for which introns were lost in *A. lycopersici*, but conserved in five other closely related or high-quality mite or insect genomes (see Materials and methods), the impact of intron loss on *A. lycopersici*-encoded protein sequences was generally minimal. In fact, in the respective protein sequences spanning 97 of 100 *A. lycopersici*-specific intron loss events (97%), multi-species alignments did not reveal insertions or deletions (indels) of amino acid residues (e.g. **Figure 2c**, and **Supplementary file 1** — 'Table S3' Tab and **Supplementary file 3**); for the remaining few cases (3%), the respective sites of loss events in *A. lycopersici* were coincident with the gain or loss of one or several amino acid residues (e.g. **Figure 2d**). Within this gene set, similar findings were apparent for the larger number of *A. lycopersici* intron losses as compared to intron sites conserved between the two closest relatives (*D. pteronyssinus* and *T. urticae*; **Supplementary file 3**). Despite striking examples of intronless genes arising from the loss of multiple conserved introns, as for *aculy03g01320* (**Figure 2c**), some *A. lycopersici* genes have both lost and retained arthropod conserved introns (i.e. *aculy02g00250*, *aculy03g02140*, and *aculy01g28080*, **Supplementary file 3**).

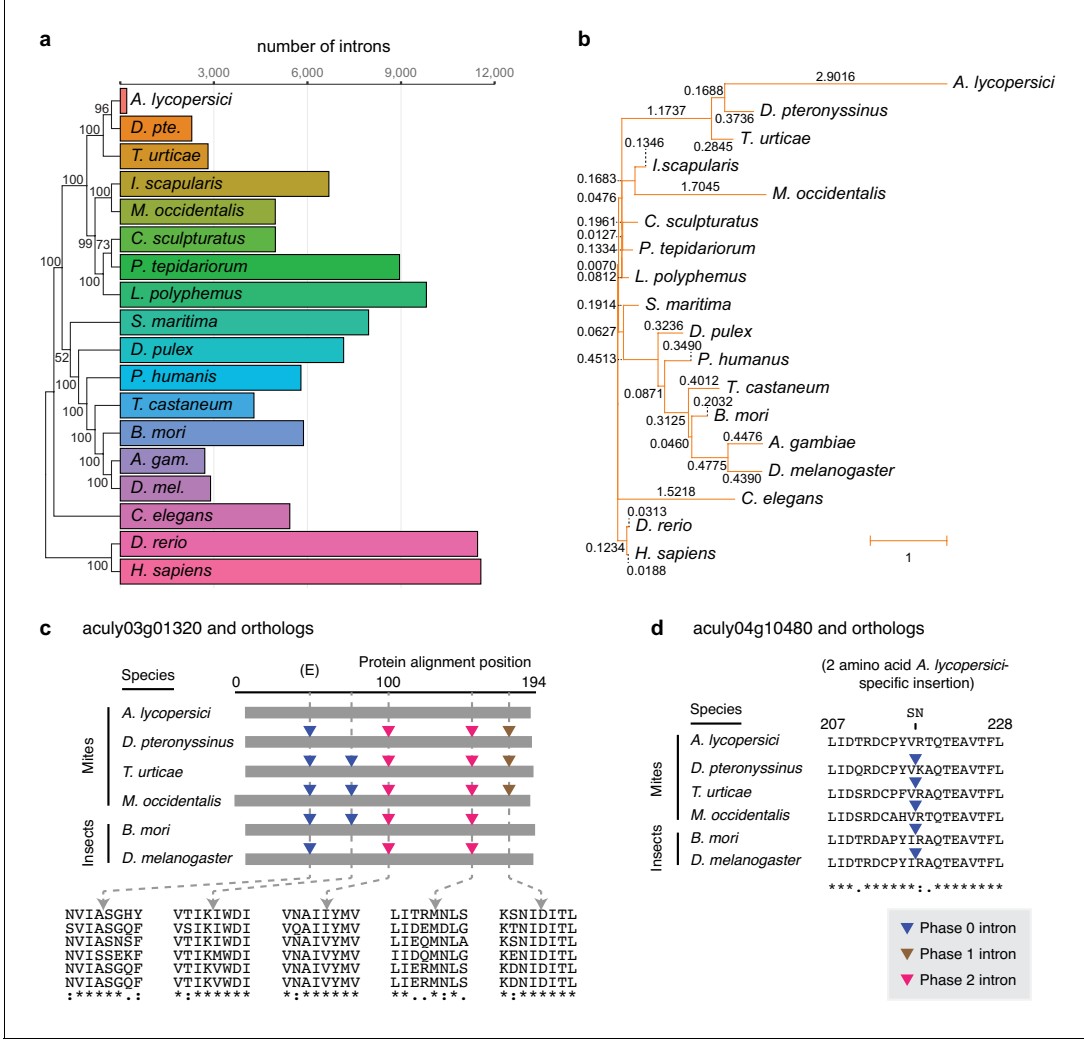

**Figure 2.** Number of conserved introns and intron loss rate across 18 metazoan species. (**a**) Phylogenetic tree built from 147 single copy orthologues (left; numbers at nodes indicate bootstrap support), and a histogram of introns present at 29,447 conserved positions identified by the software package Malin (right). (**b**) Phylogenetic tree with branch lengths labeled and scaled to the intron loss rate calculated by Malin. The unedited tree in both panels is given in *Figure 2—figure supplement 3*, and was, together with 2371 orthologous protein clusters (*Supplementary file 2*), used as input for Malin. (**c**) Alignment of *A. lycopersici* aculy03g01320 (which encodes an ADP-ribosylation factor-like 8, or Arl8, protein) with single copy orthologues from five other mite and insect species as indicated. Analogous positions of phase 0, 1, and 2 introns are denoted by colored triangles (legend, bottom right), with amino acids at the analogous intronic positions indicated beneath (identity, similarity, and non-similarity are indicated by '*', ':', and '.', respectively, for aculy03g01320 and its orthologue from *D. pteronyssinus*, the most closely related genome; in descending order, the sequence identifiers are aculy03g01320.1, g8154.t1, tetur10g00460, rna18006, BGIBMGA010943-RA, and FBtr0339723). The letter 'E' indicates that this intron position is conserved across other model organisms in Eukaryota; *Dictyostelium purpureum* (GenBank Accession XM_003283650), *C. elegans* (NM_070390.9), *H. sapiens* (NM_018184.3), *Monosiga brevicollis* (XM_001744342.1), and *A. thaliana* (NM_114847.5). (**d**) Local protein alignment, after panel c, revealing a candidate imprecise intron loss event in *aculy04g10480* (which encodes a polymerase delta-interacting protein) in *A. lycopersici* (insertion of S and N amino acid residues, top). Numbers denote positions in the *A. lycopersici* orthologue; sequence identifiers, in descending order, are aculy04g10480.1, g5664.t1, tetur01g12540, rna9399, BGIBMGA013121-RA, and FBtr0078681. Panels (**c**) and (**d**) are drawn based on Malin output. Other findings for intronic features and factors contributing to *A. lycopersici*'s genome reduction, and the supporting analyses, are presented in *Figure 2—figure supplements 1, 2, 4, 5* and *6*.

The online version of this article includes the following figure supplement(s) for figure 2:

**Figure supplement 1.** Transposable element (TE) composition of the genome of *A. lycopersici* as well as that of four other animals.
**Figure supplement 2.** Gene and TE density along the major *A. lycopersici* genome scaffolds.
**Figure supplement 3.** Maximum likelihood phylogenetic analysis of 18 metazoan species including *A. lycopersici*.
**Figure supplement 4.** Intron gain rate across 18 metazoan species including *A. lycopersici*.
**Figure supplement 5.** Density plot of conserved intron positions identified by Malin in 18 metazoan species.
**Figure supplement 6.** Median length of all introns in 18 metazoan species.

## Gene family contractions predominate in *A. lycopersici*

As revealed by the clustering algorithm implemented in the CAFE software (*Han et al., 2013*), *A. lycopersici* exhibits one of the highest rates of gene family contractions (1725), and by far the lowest rate of gene family expansions (206), among the 18 metazoans we analyzed (*Figure 3*; input data for the analysis are provided in *Supplementary file 4* and *Supplementary file 5*). It also has the lowest average expansion per gene family (*Supplementary file 1* — 'Table S4' Tab). Of the 105 gene families that were identified as 'rapidly evolving' in *A. lycopersici*, only four – as represented by orthogroups (OGs) OG0000007 (containing an Asteroid domain: IPR026832), OG0000546 (containing a Major Facilitator Superfamily, or MFS, domain: IPR011701), OG0000583 (containing a Troponin domain: IPR001978), and OG0002260 (hypothetical proteins) – were identified as expanding. The remaining 101 families were all identified as contracting (*Supplementary file 1* — 'Table S5' Tab). Six of these contracting families did contain more than 10 members in *A. lycopersici* (OG0000000, containing a Zinc finger C2H2-type domain: IPR013087; OG0000003, containing a Homeobox domain: IPR001356; OG0000005, containing a Serine protease, trypsin domain: IPR001254; OG0000014, containing a Cytochrome P450 domain: IPR001128; OG0000015, containing a G-protein-coupled receptor, rhodopsin-like domain: IPR000276; G0000025, containing a Homeobox domain: IPR001356) and, except for OG0000014 containing members of the P450 family, which is known to have only few orthologous relationships (*Feyereisen, 2011*), on average 72.2% of retained *A. lycopersici* genes had an orthologue in the majority of chelicerate species (*Supplementary file 1* — 'Table S6' Tab). Further, among the 101 rapidly contracted gene families we identified families previously implicated in mite and insect xenobiotic detoxification (*Dermauw et al., 2013a*; *Dermauw et al., 2013b*; *Snoeck et al., 2018*; *Van Leeuwen and Dermauw, 2016*) – carboxyl/choline esterases (CCEs: OG0000021 and OG0001201), cytochrome P450 monooxygenases (CYPs: OG0000014, OG0000030 and OG0000052), glutathione-S-transferases (GSTs: OG0000102, OG0000124), short-chain dehydrogenases/reductases (SDRs: OG0000096), ATP-binding cassette

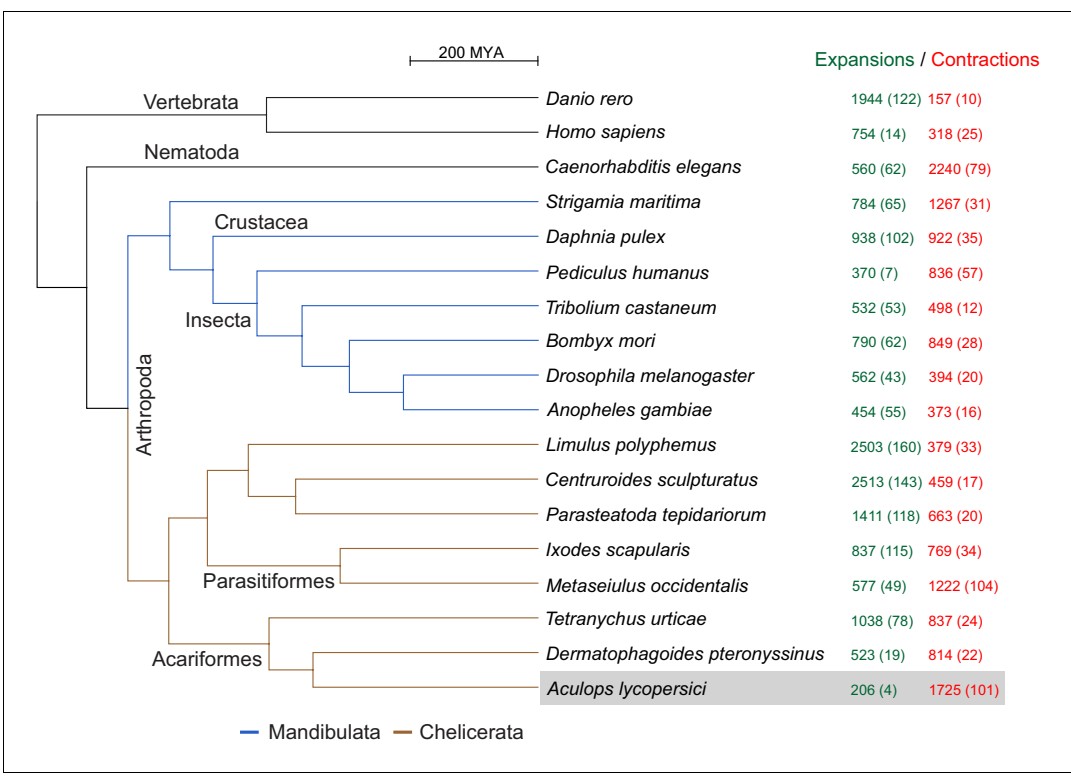

**Figure 3.** CAFE analysis of 6487 metazoan orthogroups. The number of expanding orthogroups are indicated in green font, while contracting orthogroups are indicated in red font. The number of rapidly expanding or contracting orthogroups (p-value<0.05) is shown in parentheses and details regarding these orthogroups can be found in *Supplementary file 1* — 'Table S5' Tab and 'Table S7' Tab.

(ABC) transporters (ABCs: OG0000051 and OG0000109) and MFS proteins (OG0000029, OG0000071, OG0000099, OG0000187) (*Supplementary file 1* — 'Table S5' Tab and 'Table S7' Tab). Given the role of these families in herbivory and host plant use (*Després et al., 2007*; *Heckel, 2014*; *Van Leeuwen and Dermauw, 2016*), we analyzed a selection of these gene families in detail (see the following sections).

We also found 315 orthogroups with no members in *A. lycopersici* but at least one member in all other arthropod species. This is the highest number of absent orthogroups of all arthropods included in our analysis, is ~2-fold more than those lacking in *D. pteronyssinus* (171), and more than threefold those absent in *T. urticae* (101) (*Supplementary file 1* — 'Table S8' Tab). A gene ontology (GO) enrichment analysis for *D. melanogaster* members within these conserved arthropod orthogroups without *A. lycopersici* members revealed that N-acetylglucosamine metabolic process (GO:0006044), transferase activity (GO:0016740) and Golgi apparatus (GO:000579) were the most highly significantly enriched GO terms within the Biological Process, Molecular Function and Cellular Component GO categories, respectively (*Supplementary file 1* — 'Table S9' Tab). Lastly, we found that 427 *D. melanogaster* essential genes (*Aromolaran et al., 2020*) coded for members of 390 orthogroups. Forty-eight of these essential orthogroups did not have members within the Acariformes, the mite superorder comprising *A. lycopersici*, *D. pteronyssinus,* and *T. urticae*, while 21 (5.4%) orthogroups were absent in *A. lycopersici* but present in other acariform mites (*Supplementary file 1* — 'Table S10' Tab).

Furthermore, in a number of cases, orthogroups absent in *A. lycopersici* harbor conserved genes with potential roles in the development of tissues or structures that are absent or modified in the russet mite relative to other chelicerates or insects (see also Discussion, and Results section, 'Loss of highly conserved transcription factors'). For instance, orthologues of *Drosophila* unkempt, a known developmental regulator, and *Drosophila* dachs, essential for appendage growth, are both absent in *A. lycopersici* but present in all other arthropods (OG0002898 and OG0006002, respectively). Dachs is known to interact with four-jointed (*Buckles et al., 2001*), which is also absent in *A. lycopersici*, even though it is present in all insect and chelicerate species included in our analysis (OG0003305). Finally, *fat* belongs, together with *dachs* and *four-jointed*, to the Fat/Hippo pathway and plays a key-role in tissue proliferation and development in both invertebrates and vertebrates (*Simon et al., 2010*). Although *dachsous*, another player in this pathway, is present (aculy04g02000 in OG0001018), a *fat* orthologue could not be identified in *A. lycopersici* while this orthologue was found in other acariform mites (OG0000383, *Supplementary file 1* — 'Table S7' Tab).

## Detoxification genes

We curated the *A. lycopersici* genome for sequences encoding established detoxification enzymes (*Després et al., 2007*; *Heckel, 2014*; *Van Leeuwen and Dermauw, 2016*) including GSTs, CCEs, and CYPs. In *A. lycopersici*, detoxification gene families are especially reduced, with a mere 4 GSTs, 8 CCEs, and only 23 CYPs (*Table 2*, *Figure 4a*, and *Figure 4—figure supplements 1*, *2* and *3*; *Van Leeuwen and Dermauw, 2016*). In particular, the number of GSTs and CCEs is remarkably low (see Discussion). This finding was corroborated by mining of the *A. lycopersici* transcriptome assembly (the 4 GSTs and 8 CCEs present in the genome assembly were also present in transcriptome assembly, with no other transcript contigs with homology to GSTs or CCEs identified). Of note, half of the GSTs and almost all (7 out of 8) CCE genes in *A. lycopersici* are evolutionarily conserved across chelicerates or arthropods (*Figure 4—figure supplements 1* and *2*). We also examined transporters of the ABC family and MFS proteins that have been implicated in detoxification responses in arthropod species, although transporters in both of these families have diverse other roles as well (*de la Paz Celorio-Mancera et al., 2013*; *Dermauw et al., 2013a*; *Dermauw et al., 2013b*; *Dermauw and Van Leeuwen, 2014*; *Govind et al., 2010*). In contrast to genes encoding 'classic' detoxification enzymes like CYPs, CCEs, or GSTs, dramatic reductions in ABC transporter genes were not observed. For example, *A. lycopersici* has 9 ABCC and 16 ABCG transporters, while 22 and 2 are present in *M. occidentalis* and 39 and 23 are present in *T. urticae*, respectively (*Table 2*, *Figure 4—figure supplement 4*). Further, in contrast to the trend for contractions of the classic detoxification gene families, we also observed two *A. lycopersici* expansions - comprising three orthogroups, OG0000024, OG0000546, and OG0006109 - of the MFS, which is involved in membrane-based transport of small molecules (*Figure 4b*, *Figure 4—figure supplement 5*; *Pao et al., 1998*; *Yan, 2015*).

**Table 2.** Detoxification enzyme (CYPs, GSTs, CCEs) and ABC transporter gene family size in *A. lycopersici*, *T. urticae*, *M. occidentalis*, and *D. melanogaster*.

| Detoxification enzyme | *A. lycopersici* | *T. urticae* | *M. occidentalis* | *D. melanogaster* |
|---|---|---|---|---|
| **CYPs (total)** | **23** | **78*** | **63** | **86** |
| CYP2 | 1 | 38 | 16 | 7 |
| CYP3 | 17 | 9 | 23 | 36 |
| CYP4 | 2 | 26 | 19 | 32 |
| Mito Clan | 3 | 5 | 5 | 11 |
| **GSTs (total)** | **4** | **31** | **13** | **37** |
| Delta/Epsilon | 1 | 16 | 3 | 25 |
| Mu | 2 | 12 | 5 | 0 |
| Omega | 0 | 2 | 3 | 5 |
| Sigma | 0 | 0 | 0 | 1 |
| Theta | 0 | 0 | 0 | 4 |
| Zeta | 1 | 1 | 1 | 2 |
| Unknown | 0 | 0 | 1 | 0 |
| **CCEs (total)** | **8** | **69** | **44** | **35** |
| Dietary class (clade A-C) | 0 | 0 | 0 | 13 |
| Hormone class | | | | |
| D (integument CCEs) | 0 | 0 | 0 | 3 |
| E (secreted beta-esterases) | 0 | 0 | 0 | 2 |
| F (dipteran JHEs[†]) | 0 | 0 | 0 | 3 |
| F' (chelicerate JHEs) | 0 | 2 | 1 | 0 |
| Neurodevelopmental class | | | | |
| H (glutactins) | 0 | 0 | 0 | 4 |
| J (AChE) | 1 | 1 | 1 | 1 |
| J' (Acari-specific CCEs) | 0 | 32 | 19 | 0 |
| J'' (Acari-specific CCEs) | 0 | 22 | 15 | 0 |
| K (gliotactin) | 1 | 1 | 1 | 1 |
| L (neuroligins) | 2 | 5 | 5 | 4 |
| M (neurotactin) | 1 | 1 | 0 | 1 |
| U (unchar. conserv. clade in Acariformes/*L. polyphemus*) | 2 | 3 | 0 | 0 |
| I (unchar. conserv. clade in insects) | 0 | 0 | 0 | 2 |
| No clear clade assignment | 1 | 2 | 2 | 1 |
| **ABCs (total)** | **44** | **103** | **55** | **56** |
| ABCA | 4 | 9 | 8 | 10 |
| ABCB-FT[‡] | 3 | 2 | 1 | 4 |
| ABCB-HT[§] | 1 | 2 | 4 | 4 |
| ABCC | 9 | 39 | 22 | 14 |
| ABCD | 2 | 2 | 4 | 2 |
| ABCE | 1 | 1 | 1 | 1 |
| ABCF | 3 | 3 | 3 | 3 |
| ABCG | 16 | 23 | 2 | 15 |
| ABCH | 5 | 22 | 6 | 3 |
| Unknown | 0 | 0 | 4 | 0 |
| **Total** | **79** | **281** | **175** | **214** |

Numbers and class/clade/subfamily assignments were derived from previous studies (*Grbić et al., 2011*; *Wei et al., 2020*; *Wu and Hoy, 2016*) and this study.

*Of the 81 *T. urticae* CYPs identified by *Grbić et al., 2011*, three CYP genes (*tetur46g00150, tetur46g00170 and tetur47g00090*) and *tetur602g00010* were considered as allelic variants and a pseudogene, respectively, and one new full-length CYP gene (*tetur01g13730*) was identified in this study.

†JHE, juvenile hormone esterases.

‡FT, full transporter.

§HT, half transporter.

## Chemosensory and related receptors

To see if *A. lycopersici*'s specialized lifestyle has had a notable impact on chemoreception, we also exhaustively mined and annotated the *A. lycopersici* genome for gustatory receptors (GRs), degenerin/epithelial Na+ channels (ENaCs), ionotropic receptors (IR) and transient receptor potential (TRP) channels. Members of these four families have been previously documented to play important roles in sensing environmental (chemical) cues in other arthropod species (*Damann et al., 2008*; *Hoy et al., 2016*; *Ngoc et al., 2016*; *Robertson et al., 2003*; *Rytz et al., 2013*; *Whiteman and Pierce, 2008*). The GR family, which contains seven transmembrane spanning regions (*Touhara and Vosshall, 2009*) and is linked to the detection of sweet and bitter compounds (*Silbering and Benton, 2010*), was the most strongly reduced, with only two of these genes identified (*Figure 4c*, *Figure 4—figure supplement 6*), as opposed to the 447 intact GRs reported in *T. urticae* (*Ngoc et al., 2016*). Further, only four ENaCs are present in the *A. lycopersici* genome (*Figure 4d*, *Figure 4—figure supplement 7*). Members of this family have recently been shown or suggested to be chemoreceptors for diverse compounds in insects and mites, but some family members likely have highly conserved roles in acid sensing (*Ben-Shahar, 2011*; *Silbering and Benton, 2010*), as well as in the perception of mechanical or osmotic cues (*Ben-Shahar, 2011*; *Zelle et al., 2013*). Of the two ENaCs likely to play these conserved roles in *T. urticae,* one is in a well-supported clade with a single ENaC in the tomato russet mite (aculy04g09940) (*Figure 4* , *Figure 4—figure supplement 7*).

The IR family, which has been linked to odorant detection (*Joseph and Carlson, 2015*), humidity and temperature sensing in *D. melanogaster* (*Enjin et al., 2016*), is markedly reduced in *A. lycopersici* compared to most insects and *M. occidentalis* (*Hoy et al., 2016*). However, the numbers are similar to those in *T. urticae* (each has four putative IRs with strong bootstrap support), including homologues of the highly conserved IR25a and IR93a receptors (*Figure 4—figure supplement 8*). Interestingly, *A. lycopersici* may have as few as six ionotropic glutamate receptors (iGluRs), as compared to 14 in *T. urticae* (*Figure 4—figure supplement 8*); proteins in this family are related to IRs, but have ultra-conserved roles in synaptic transmission in animals (*Benton et al., 2009*).

Finally, we found both expansions and contractions of the TRP family (*Figure 4—figure supplement 9*). Like the other sequenced herbivorous mite, *T. urticae*, no orthologue of TRPA1 was located, but orthologues for TRPgamma, NopmC, and TRPML are present, with three copies of NopmC as compared to *T. urticae*'s two. Unlike *T. urticae*, members of the TRPP and TRPM clades were completely absent in the russet mite, but strikingly, two putative members of the TRPV clade (Inactive and Nanchung), previously thought to be lost in mites and ticks (*Peng et al., 2015*; *Regier et al., 2010*), appear to be present.

## Loss of highly conserved transcription factors

Among two vertebrates, one nematode and the 15 arthropod species we analyzed, *A. lycopersici* has the lowest number (364) of transcription factor (TF) genes (*Supplementary file 1* — 'Table S11' Tab). Nevertheless, when accounting for the total number of genes by species, the TF fraction in *A. lycopersici* (3.55%) is higher than that of *T. urticae* (2.98%), and is within the range reported for metazoan animals (4.7% ±1.4, *Charoensawan et al., 2010*). However, a lower number of the PFAM TF domains Zinc finger (zf-C2H2 and zf-CCHC), Forkhead, Homeobox, Hormone (nuclear) receptor, HLH, bZIP_2 and T-box were found in *A. lycopersici* compared to all other species included in our analysis (*Supplementary file 1* — 'Table S11' Tab). In addition, *A. lycopersici* orthologues of the Hairy Orange protein family (hey, cwo and deadpan) have lost the Hairy Orange domain (*Figure 4—figure supplement 10*), while an orthologue of *D. melanogaster* SoxNeuro could not be identified in *A. lycopersici* despite being present in the spider and Acari genomes examined (*Figure 4—figure*

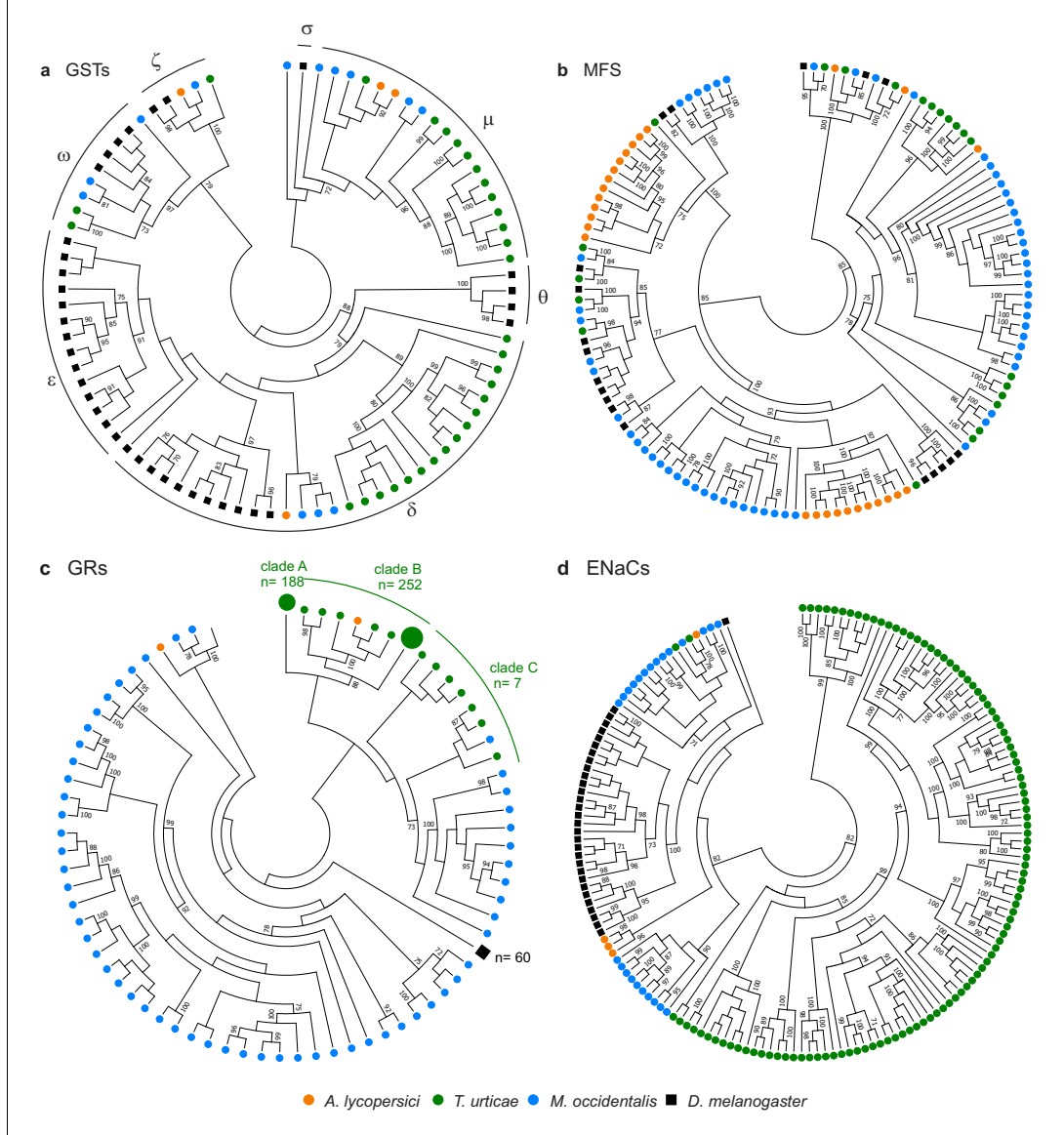

**Figure 4.** Gene family contractions and mini-expansions in *A. lycopersici*. Maximum likelihood phylogenetic analysis of selected detoxification and chemosensory families among *A. lycopersici*, *T. urticae*, *M. occidentalis* and *D. melanogaster*. (**a**) Glutathione-S-transferases (GSTs); the different GST classes (zeta, theta, delta, epsilon, omega, mu, sigma) are indicated with arches. (**b**) Major facilitator superfamily (MFS). (**c**) Gustatory receptors (GRs). (**d**) Epithelial Na+ Channels (ENaCs). All trees are midpoint rooted and only topology is shown. Gustatory receptors for *D. melanogaster* as well as the species-specific class A and B expansions identified in *T. urticae* are collapsed for clarity. Only bootstrap values above 70 are shown. Phylogenetic reconstructions for gene families, or analyses of domain losses in *A. lycopersici* in arthropod conserved genes, are given in *Figure 4—figure supplements 1–20*. For panels a-d, the detailed versions for each tree, including sequence identifiers, can be found in *Figure 4—figure supplements 1*, *5*, *6* and *7*, respectively. The alignments used for phylogenetic inference can be found in *Supplementary file 7*.

The online version of this article includes the following figure supplement(s) for figure 4:

**Figure supplement 1.** Phylogenetic analysis of GST protein sequences of *A. lycopersici*.

**Figure supplement 2.** Phylogenetic analysis of CCE protein sequences of *A. lycopersici*.

**Figure supplement 3.** Phylogenetic analysis of CYP protein sequences of *A. lycopersici*.

**Figure supplement 4.** Phylogenetic analysis of nucleotide-binding domains of ABC proteins of *A. lycopersici*.

**Figure supplement 5.** Phylogenetic analysis of MFS protein sequences of *A. lycopersici*.

**Figure supplement 6.** Phylogenetic analysis of GRs of *A. lycopersici*.

**Figure supplement 7.** Phylogenetic analysis of ENaCs of *A. lycopersici*.

**Figure supplement 8.** Phylogenetic analysis of ionotropic and related receptors of *A. lycopersici*.

**Figure supplement 9.** Phylogenetic analysis of TRP channels of *A. lycopersici*.

*Figure 4 continued on next page*

*Figure 4 continued*

**Figure supplement 10.** Alignment of the Hairy Orange domain region from deadpan, hey and cwo proteins of *A. lycopersici*, *D. pteronyssinus*, and *T. urticae* with deadpan, hey and cwo of *D. melanogaster*.

**Figure supplement 11.** Bayesian phylogenetic analysis of *A. lycopersici* Sox proteins.

**Figure supplement 12.** Alignment of the DNA-binding domain and the ligand-binding domain region of *D. melanogaster* E75 with those of *T. urticae*, *D. pteronyssinus* and *A. lycopersici*.

**Figure supplement 13.** Alignment of the DNA-binding domain and the ligand-binding domain region of *D. melanogaster* HR4 with those of *T. urticae* and *A. lycopersici*.

**Figure supplement 14.** Alignment of the DNA-binding domain and the ligand-binding domain region of *D. melanogaster* HR38 with those of *T. urticae*, *D. pteronyssinus* and *A. lycopersici*.

**Figure supplement 15.** Alignment of the DNA-binding domain and the ligand-binding domain region of *D. melanogaster* HR51 with those of *T. urticae*, *D. pteronyssinus*, and *A. lycopersici*.

**Figure supplement 16.** Alignment of the DNA-binding domain and the ligand-binding domain region of *D. melanogaster* SVP with those of *T. urticae*, *D. pteronyssinus*, and *A. lycopersici*.

**Figure supplement 17.** Alignment of the DNA-binding domain and the ligand-binding domain region of *D. melanogaster* DSF with those of *T. urticae*, *D. pteronyssinus*, and *A. lycopersici*.

**Figure supplement 18.** Phylogenetic analysis of *A. lycopersici* protein sequences with a T-box (PF00907) domain.

**Figure supplement 19.** Phylogenetic analysis of *A. lycopersici* UGT protein sequences.

**Figure supplement 20.** Phylogenetic analysis of *A. lycopersici* C1A proteases.

*supplement 11*). Among nuclear receptors (NRs), we identified eight canonical NRs in the *A. lycopersici* genome (E78, HR3, EcR, two RXRs, ERR, FTZ-F1, HR96) that contained both a DNA-binding domain (DBD) and a ligand-binding domain (LBD). However, no homologues of the evolutionary conserved NRs HNF4, HR39, HR78, and HR83 (*Bodofsky et al., 2017*; *Bonneton and Laudet, 2012*), nor a homologue of the *T. urticae* Photoreceptor-specific NR (PNR), were detected in the *A. lycopersici* genome, even though HR78, HNF4, and PNR are present in *D. pteronyssinus* (*Supplementary file 1* — 'Table S12.1' Tab and 'Table S12.2' Tab). Further, for six nuclear receptors (E75, DSF, HR4, HR38, HR51, and SVP) that are evolutionary conserved across arthropods and normally have a canonical (DBD+LBD) structure (*Fahrbach et al., 2012*; *Grbić et al., 2011*; *Hwang et al., 2014*; *Litoff et al., 2014*), an LBD was not predicted for the respective *A. lycopersici* homologues. LBDs for all of these except HR4 were predicted for both the *D. pteronyssinus* and *T. urticae* homologues (*Supplementary file 1* — 'Table S12.1' Tab and 'Table S12.2' Tab, *Figure 4— figure supplements 12–17*).

The basic helix-loop-helix (bHLH) gene family is an ancient family found in fungi, plants, and animals, and members of this family are essential both for organisms to respond to environmental factors, as well as for cellular differentiation during development (*Skinner et al., 2010*). The *D. melanogaster achaete* and *scute* bHLH genes play crucial roles in bristle development (*García-Bellido and de Celis, 2009*). Within the bHLH family group we found that *T. urticae*, *M. occidentalis* and *I. scapularis* have five bHLH proteins with an achaete-scute InterPro domain (IPR015660), while only three were found in both *D. pteronyssinus* (g4111.t1, g7028.t1 and g6164.t1) and *A. lycopersici* (aculy01g18470, aculy01g18540 and aculy02g28230).

A number of other specific transcription factors that are highly conserved among most arthropods are also absent from the *A. lycopersici* genome. For *A. lycopersici*, we were unable to identify *proboscipedia*, a member of the Hox gene family. Members of this family (*labial*, *proboscipedia*, *Hox3/ zen*, *Deformed*, *Sex combs reduced*, *fushi tarazu*, *Antennapedia*, *Ultrabithorax*, *abdominal-A*, and *Abdominal-B*) encode homeodomain transcription factors and act to determine the identity of segments along the anterior–posterior axis in arthropods (*Hughes and Kaufman, 2002*). *Proboscipedia* is present in all chelicerate genomes (horseshoe crab, scorpions, spiders, mites and ticks) for which Hox genes have been analyzed (*Figure 5*, *Supplementary file 1* — 'Table S13.1' Tab and 'Table S13.2' Tab, *Supplementary file 6*; *Di et al., 2015*; *Hoy et al., 2016*; *Kenny et al., 2016*; *Schwager et al., 2017*), and is believed to be ancestral to all arthropods (*Pace et al., 2016*). Of particular note, *proboscipedia* is located in close proximity (<35 kb) of *labial* in Acariformes, but in *Aculops labial* was the only Hox gene that was present on scaffold 2 (*Supplementary file 1* — 'Table S14' Tab). Furthermore, *A. lycopersici* lacks a homologue of the T-box encoding gene *org-1* (*Figure 4—figure supplement 18*), which in *D. melanogaster* plays a pivotal role in diversification of circular visceral muscle (*Schaub and Frasch, 2013*). Finally, we also mined the *A. lycopersici* genome

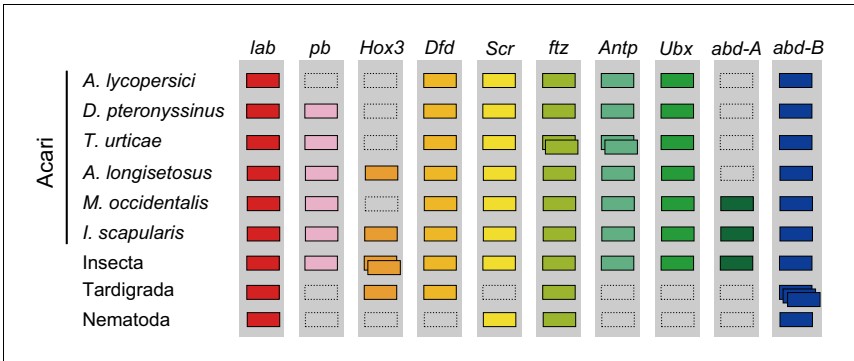

**Figure 5.** Hox genes in Acari and other ecdysozoan lineages. Hox orthology groups are indicated by different colored boxes. Gray boxes with a dashed outline represent missing Hox genes. Some species have duplications of Hox genes and these are indicated by multiple boxes that overlap. *T. castaneum*, *H. dujardini* and *C. elegans* were selected as representative species for the Hox gene clusters of Insecta, Tardigrada and Nematoda, respectively.

for transcription factors and other genes involved in circadian rhythm (so-called 'clock' genes) (*Supplementary file 1* — 'Table S15' Tab). Orthologues of the helix-loop-helix TFs *cycle*, *Clock* and *tango* and the bZIP TF *vrille* were identified in the *A. lycopersici* genome. However, we did not identify *period* and *timeless*, known negative regulators of *Clock* and *cycle* (*Lee et al., 1999*; *Peschel and Helfrich-Förster, 2011*). Other circadian regulators, like the circadian photoreceptor *cryptochrome* and the bZIP TF *PAR-domain protein 1ε*, were also not identified, even though these are present in *T. urticae* (*Hoy et al., 2016*).

## Horizontally transferred genes

We identified 18 putatively intact horizontal gene transfer (HGT) candidate genes (*Supplementary file 1* — 'Table S16' Tab), and performed subsequent phylogenetic analyses that suggested that nine were acquired from a foreign source. Seven of these genes code for UDP-glyco-syltransferases (UGTs), members of which have well documented roles in xenobiotic detoxification (*Snoeck et al., 2019*). Phylogenetic inference with all *T. urticae*, *D. pteronyssinus* and *A. lycopersici* UGTs (80, 27, and 7, respectively) indicated that the seven UGTs in the tomato russet mite genome were the result of a lineage-specific expansion (*Figure 4—figure supplement 19*). Although we did not observe a clear phylogenetic signature of HGT (*Wybouw et al., 2016*), our phylogenetic reconstruction is consistent with previous studies which indicated that, prior to the formation of the Acariformes lineage, an ancestral mite species laterally acquired a *UGT* gene copy from a bacterial source (*Ahn et al., 2014*)(*Wybouw et al., 2018*).

Two intact genes of bacterial origin (*aculy01g38350* and *aculy04g02470*) were also identified in the tomato russet mite genome that are predicted to code for enzymes in the microbial and plant pantothenate biosynthesis pathway (an apparent duplicate of *aculy01g38350* was also uncovered, but the coding sequence was disrupted, and it lacked expression, suggesting it is a pseudogene) (*Figure 6*). PCR amplification linked both laterally acquired genes with either neighboring intron-containing genes (*aculy01g38350*) or conserved eukaryotic genes (*aculy04g02470* is located next to *aculy04g02480*, which encodes a Gtr1/RagA protein); in addition, an *aculy01g38350* transcript (Illumina contig 1934) had a polyA tail, suggestive of eukaryotic transcription (*Figure 6—figure supplement 1*). Pantothenate, or vitamin B5, is a life-essential compound, and whereas plants and bacteria are able to synthesize this compound de novo, animals rely on dietary uptake. Genes for pantothenate synthesis are present in tetranychid mites, and genomic and phylogenetic approaches have pointed to an ancient HGT event prior to speciation within the Tetranychidae family for both genes. Constrained tree tests rejected the topology where ketopantoate hydroxymethyltransferase of *A. lycopersici* was the sister lineage to the group of spider mite biosynthetic proteins, but not for pantoate β-alanine ligase, suggesting that *A. lycopersici* acquired the ketopantoate hydroxymethyltrans-ferase gene from a different bacterial donor species (*Figure 6*, Approximately Unbiased tests, p-value cut-off of 0.01).

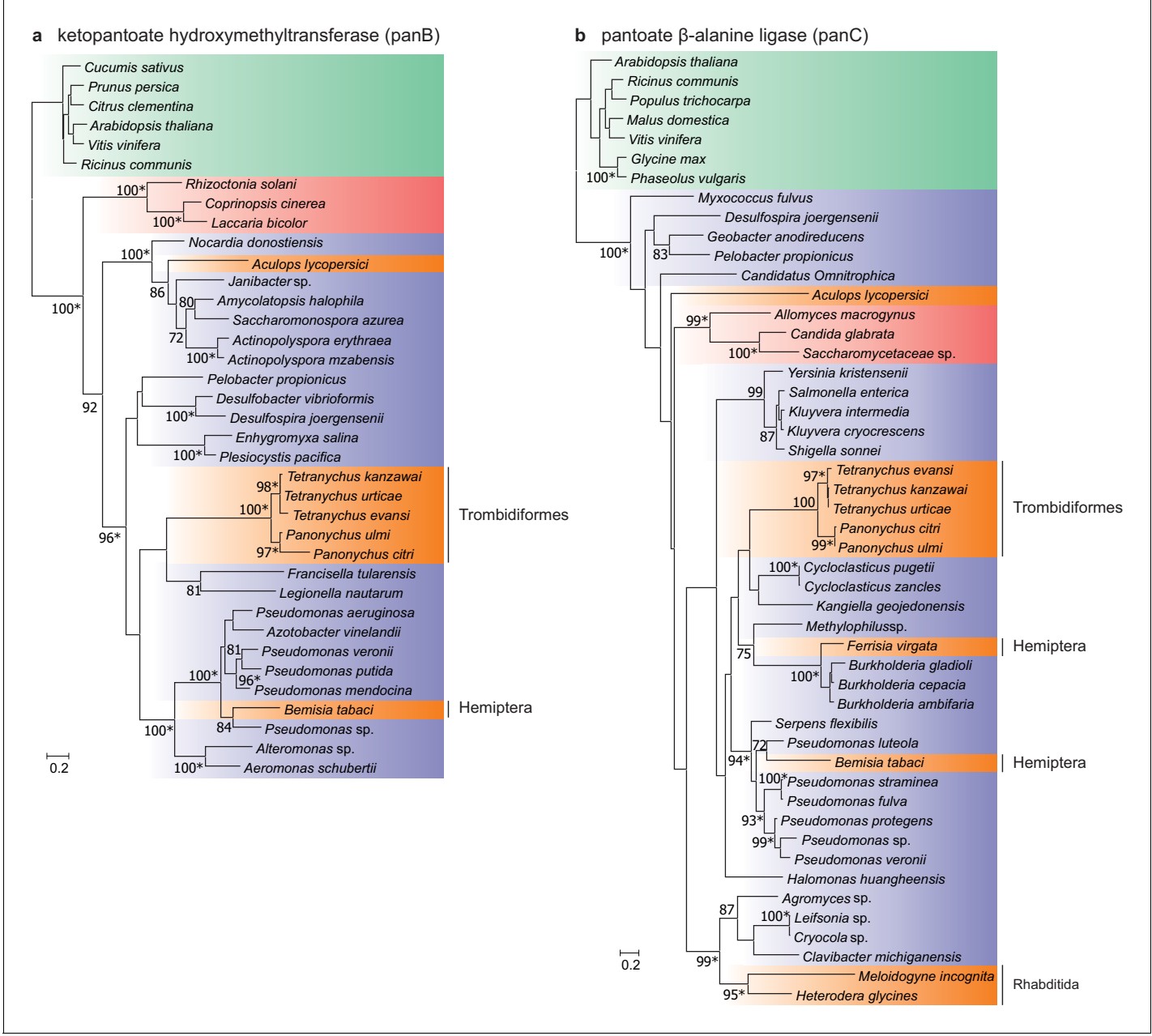

**Figure 6.** Maximum-likelihood phylogenetic inference for ketopantoate hydroxymethyltransferase and pantoate β-alanine ligase of *A. lycopersici*. (**a**) Ketopantoate hydroxymethyltransferase. (**b**) Pantoate β-alanine ligase. Branches are color coded depending on their position within the tree of life; plants: green, animals: orange, fungi: red and bacteria: blue. RAxML phylogenetic reconstructions are consistent with the evolutionary scenario of independent horizontal transfer events of the two pantothenate biosynthetic genes in the *A. lycopersici* lineage, tetranychid spider mites, and hemipterans. Only RAxML bootstrap support values higher than 70 are depicted and the scale bars represent 0.2 amino acid substitutions per site. Informative nodes were identical and well-supported in another maximum-likelihood analysis (IQ-TREE; an asterisk indicates nodes with ultrafast bootstrap values above or equal to 95 in the IQ-TREE analyses). Plant homologues were used to root both phylogenetic trees. The alignments used for phylogenetic inference can be found in *Supplementary file 7*.

The online version of this article includes the following figure supplement(s) for figure 6:

**Figure supplement 1.** Integration of the *ketopantoate hydroxymethyltransferase* (*aculy01g38350*, *panB*) and *pantoate β-alanine ligase* (*aculy04g02470*, *panC*) genes into the *A. lycopersici* genome.

In *T. urticae* the acquisition of pantothenate biosynthetic genes is accompanied by the horizontal gene transfer of two methylenetetrahydrofolate dehydrogenases (MTHFDs), enzymes of the folate pathway and connected to the pantothenate biosynthesis pathway (*Wybouw et al., 2018*). Although such a HGT was not detected in *A. lycopersici*, an expansion of MTHFDs was detected compared to other mite species (OG0000706 in *Supplementary file 1* — 'Table S7' Tab).

## Secreted proteins

Small molecules or proteins produced in salivary glands are one mechanism by which arthropod herbivores can manipulate the defenses of their host plants. As *A. lycopersici* is able to potently suppress tomato defenses (*Glas et al., 2014*; *Schimmel et al., 2018*), we predicted its secretome, and found that 612 of the 10,263 annotated *A. lycopersici* proteins (6%) are putatively secreted (*Supplementary file 1* — 'Table S17' Tab). Only one of the more than 600 secreted *A. lycopersici* proteins (aculy02g17370, a glycosyl hydrolase, family 13, IPR013780) had a best BLASTp hit with a *T. urticae* protein that was previously identified in *T. urticae* saliva using an LC-MS/MS (*Jonckheere et al., 2016*). More than half (351) of these proteins were absent in orthogroups in non-herbivorous arthropod species, and are less than 350 amino acids in length. Only 15 of these 351 proteins belonged to an orthogroup with more than one member in *A. lycopersici* (OG0006384, OG0009325, OG0009954 and OG0010904). Among these, OG0009325 contains three short *A. lycopersici* proteins <90 amino acids in length (aculy01g11450, aculy01g12600, and aculy01g12690). Of note, the gene encoding the single *T. urticae* representative in this group, *tetur24g01070*, was previously found to be overexpressed in the *T. urticae* salivary gland region (*Jonckheere et al., 2016*). OG0006384, on the other hand, contains cysteine peptidases (Peptidase C1A, papain C-terminal domain; InterPro IPR000668), which are enzymes reported to have key roles in plant-pathogen/pest interactions (*Shindo and Van der Hoorn, 2008*), and for which two lineage-specific expansions are present in *A. lycopersici* (*Figure 4—figure supplement 20*).

## Small RNA pathways

We also characterized components of small RNA pathways that might be of potential relevance for agricultural control methods. The *A. lycopersici* genome harbors highly conserved miRNA sequences, such as *let-7*, *miR-1*, and *miR-9a* (*Supplementary file 1* — 'Table S18' Tab). However, in contrast to *T. urticae*, a clear *A. lycopersici* homologue of Exportin-5, a dsRNA-binding protein mediating nuclear transport of pre-miRNAs (*Bohnsack et al., 2004*; *Kim, 2005*), is lacking, suggesting a deviating miRNA pathway in *A. lycopersici*. In line with the latter hypothesis, we could not identify an *A. lycopersici* homologue of *Staufen*, while this gene is present in *T. urticae* (*Grbić et al., 2011*; *Supplementary file 1* — 'Table S19' Tab) and other arachnids (OrthoDb v 9.1, group EOG091G07A0 and EOG090Z04UZ, respectively) and was shown to negatively modulate miRNA activity in the nematode *C. elegans* (*Ren et al., 2016*).

The *A. lycopersici* genome contains, in line with *T. urticae*, clear homologues of Dicer-1, Loquacious, Drosha and Pasha and an expansion of the AGO1 and PIWI/AGO3 subfamilies. Of note, we found one *A. lycopersici* protein (aculy02g00240) that was highly homologous to the *T. castaneum* Dicer-1 enzyme (bitscore of 294) and that contained both an RNA-binding domain (PAZ-domain, cl00301) and the RNAse III domain (cd00593) while two *A. lycopersici* proteins (aculy02g04810 and aculy02g19970) showed reciprocal BLASTp hits with *T. castaneum* Dicer-2 and Dicer-1, respectively, but were relatively short (about 500 amino acids (aa) compared to 1726 aa for aculy02g00240) and only contained the RNAse III domain. However, the genes encoding these proteins are located next to a sequencing gap in the current assembly and it could be that gene-models for these Dicer-like enzymes are not complete. Similar to *T. urticae*, we could not identify clear homologues of R2D2 and AGO2 (*Grbić et al., 2011*; *Supplementary file 1* — 'Table S19' Tab), suggesting that the siRNA pathway is either absent or non-canonical in both mite species (*Okamura et al., 2011*).

Further, important players in the PIWI-interacting RNA (piRNA) pathway (*Iwasaki et al., 2015*) were identified in the *A. lycopersici* genome (PIWI/AGO3, Zucchini, Armitage, Maelstrom and SoYb; *Supplementary file 1* — 'Table S19' Tab), while homologues of Armitage and Zucchini could not be identified in *T. urticae*, which is in line with the recently suggested non-canonical piRNA pathway in *T. urticae* (*Supplementary file 1* — 'Table S19' Tab, *Huang et al., 2014*; *Mondal et al., 2018b*).

Finally, RNA-dependent polymerases are known to be essential for the amplification of the RNA silencing effect (systemic RNAi) in *C. elegans* and some plants (*Tomoyasu et al., 2008*). Genes encoding these enzymes are absent in insect genomes while 1 to 5 have been reported in Acari genomes (*Grbić et al., 2011*; *Hoy et al., 2016*; *Joga et al., 2016*; *Mondal et al., 2018a*; *Zong et al., 2009*). Surprisingly, we could not identify RNA-dependent polymerase genes in the *A. lycopersici* genome (*Supplementary file 1 — 'Table S19' Tab*), which might indicate that these genes have been lost since the divergence of Eriophyoidea from other acariform lineages. However, as systemic RNAi does seem to occur in some insect orders, for example, Coleoptera (*Joga et al., 2016*), we cannot exclude that systemic RNAi might also occur in *A. lycopersici*.

## Discussion

Genome size varies enormously within the Acari. While tick genomes can be larger than 2 Gb (*Gulia-Nuss et al., 2016*), those of mite species belonging to the Acariformes are small (*Gregory and Young, 2020*). This is especially true for mites within the order Sarcoptiformes, including dust mites and scabies mites, for which genomes of lengths ~55-60 Mb have been reported (*Chan et al., 2015*; *Rider et al., 2015*). Eriophyoid mites like *A. lycopersici* have traditionally been placed in the order of the Trombidiformes, but recent work suggests they belong to the Sarcoptiformes, or a sister taxon (*Arribas et al., 2020*; *Bolton et al., 2017*; *Klimov et al., 2018*; *Xue et al., 2017*). Our work supports this conjecture, as within Acariformes, *A. lycopersici* fell in a well-supported clade with the house dust mite *D. pteronyssinus* (Sarcoptiformes), as opposed to *T. urticae* (Trombidiformes) (*Figure 2—figure supplement 3*).

Mirroring that of its closest sequenced relatives, the genome of *A. lycopersici* is tiny. At 32.5 Mb, it is the smallest reported to date for an arthropod and among the smallest metazoan genomes sequenced so far (*Slyusarev et al., 2020*). Its size is also consistent with cytological data that eriophyoid mites have few chromosomes that are extremely small (*Helle and Wysoki, 1996*; *Helle and Wysoki, 1983*) and with several trends. In broad terms eukaryotic genome sizes correlate positively with larger cell (nuclei) sizes, and vary inversely with cell division times (*Elliott and Gregory, 2015*; and references therein). While little is known about the minimal cell sizes for *A. lycopersici*, the whole mite is smaller than many single eukaryotic cells and neuron somata sizes of less than 1 µm have been observed for another eriophyoid mite of similar size (*Whitmoyer et al., 1972*). *A. lycopersici* is also half the size (or less) of mites like *D. pteronyssinus*, and its minute physical stature and genome size are consistent with a recent analysis that revealed a positive correlation within Acari between organismal size and haploid DNA content (*Gregory and Young, 2020*). The *A. lycopersici* generation time, a potential (albeit imperfect) proxy for cell cycle progression, is also near the minimum reported for other mites, or for microinsects (*Danks, 2006*; *Kawai and Haque, 2004*; *Rice and Strong, 1962*). The force(s) that have led to the small physical and genome size of *A. lycopersici* are not known. However, russet mites can use their short stylets only to feed on plant epidermal cells (*Royalty and Perring, 1988*). This is in contrast to many other (larger) herbivores, including other herbivorous mites like *T. urticae* (*Bensoussan et al., 2016*), that can reach and consume the photosynthetically active, sugar-rich mesophyll cells (*Borsuk and Brodersen, 2019*; *Koroleva et al., 2000*) underneath the epidermis. The nutrient-poor diet of *A. lycopersici* may favor small physical size, and under some conditions, nutrient limitations have been proposed to select specifically for low DNA content (*Hessen et al., 2010a*). Regardless, the rapid generation time of *A. lycopersici* facilitates dense populations on its host (*Figure 1c,d*), and outcrossing by deposition of spermatophores (*Al-Azzazy and Alhewairini, 2018*) in the environment may approximate panmixia, and hence high effective population sizes, and therefore more efficient selection against the accumulation of non-coding sequences associated with large eukaryotic genomes (*Lynch et al., 2011*). Therefore, a collection of life history features may underlie the streamlining observed in the *A. lycopersici* genome.

In addition to a very low content of repetitive sequences, a derived genomic organization underpins the reduced *A. lycopersici* genome. As compared to the ~3 fold larger *T. urticae* genome (*Grbić et al., 2011*), the relative intergenic and intronic fractions are reduced, while compared to the ~2 fold larger *D. pteronyssinus* genome (*Waldron et al., 2017*), the intergenic fraction is nearly identical, while the genomic percent in introns is less. The latter reduction reflects massive intron loss in *A. lycopersici*, as 83.7% of genes were intronless, a value more than threefold higher than for

*T. urticae* or *D. pteronyssinus.* As observed in other intron-poor species (*Mourier and Jeffares, 2003*), we observed greater retention of 5′ introns in *A. lycopersici*, potentially a consequence of intron loss via 3′-biased intron removal by recombination with cDNAs following reverse transcription of spliced transcripts (also known as Reverse Transcriptase-Mediated Intron Loss, or RTMIL) (*Mourier and Jeffares, 2003*; *Roy and Gilbert, 2005*). Alternatively, or in concert, the pattern may reflect retention of 5′ introns rich in cis regulatory sequences (*Roy and Gilbert, 2005*), an explanation consistent with *A. lycopersici*'s relatively long median intron lengths as compared to other insects and mites with compact genomes (*Table 1*, *Figure 2—figure supplement 6*). Previously, comparisons of intron loss events among close relatives, where few mutational steps have occurred, have been important in establishing plausible mechanisms of intron loss (*Yenerall et al., 2011*; *Zhu and Niu, 2013*). Such analyses are challenging to perform for *A. lycopersici*, as the time of divergence from the most recent common ancestor with a sequenced genome is hundreds of millions of years. Nevertheless, for a set of *A. lycopersici* intron losses in highly conserved genes – for which confident assignment of intron positions could be made in multi-species protein alignments – the overwhelming majority of loss events were consistent with precise intron excisions (i.e. *Figure 2c*). This pattern is consistent with a major role for intron removal via RTMIL, which has also been suggested to be a frequent mechanism underlying intron loss events in the genomes of (comparatively) closely related *Drosophila* species (*Yenerall et al., 2011*). However, a more prominent role for precise (or nearly precise) genomic deletions of introns as a loss mechanism in *A. lycopersici* cannot be ruled out, especially as our analysis necessarily involved conserved genes for which imprecise intronic deletions would likely be highly detrimental. *A. lycopersici* also has a very rapid generation time, and as it is evolutionary distant from its closest sequenced relatives (*Figure 3*), many lineage-specific uncommon mutation events (such as genomic deletion of introns) have potentially been sampled. Currently, more closely related genomes are needed to distinguish between RTMIL or genomic deletions as the predominant driver of intron loss in *A. lycopersici*, as well as to assess contributions of other possible mechanisms – for instance, retrotransposition by target-primed reverse transcription of spliced transcripts (*Cordaux and Batzer, 2009*; *Wang et al., 2014*), with subsequent loss of source, intron-containing loci. Likewise, more closely related genomes will be critical to establish the timing of intron losses. As additional genomes in this lineage become available, eriophyoid mites promise to be an attractive system to investigate the dynamics of intron evolution.

Apart from the dearth of introns, the complement of coding genes in the *A. lycopersici* genome deviates from that of relatives with larger genomes, and seems to be associated with its reduced morphology and distinct life history (*Lindquist and Oldfield, 1996*). Compared to other arthropods, a mere handful of gene families were expanded, including one that encodes a troponin domain. While this result was unexpected, as troponin performs a conserved role in muscle contraction and is single or low copy number in most arthropods, in a transcriptome assembly of *Aceria tosichella*, a non-galling eriophyoid pest of wheat and other grasses, an expansion of troponin-encoding genes was also observed (*Gupta et al., 2019*). Possibly, this expansion may be related to the derived body musculature of eriophyoids, as their skeletal and peripheral musculature is very pronounced, with the latter enabling the maintenance of body turgidity (*Nuzzaci and Alberti, 1996*). Nevertheless, the dominant force in shaping the genic composition of *A. lycopersici* is loss, including for genes involved in highly conserved metazoan or arthropod cell processes (e.g. for the Golgi apparatus), as well as gene families and specific genes (or conserved domains) involved in many aspects of arthropod development and physiology. The latter include Hairy Orange domain proteins, nuclear receptors, and other transcription factors that have broadly conserved roles in animal development (*Holland, 2013*; *Iso et al., 2003*; *Pflugfelder et al., 2017*; *Sebé-Pedrós and Ruiz-Trillo, 2017*; *Shimeld et al., 2010*), and whose reduction (or simplification by domain loss) in *A. lycopersici* may be related to the eriophyoid body plan. For example, in contrast to other mites, *A. lycopersici* has no orthologue of the T-box gene *org-1*, which in *D. melanogaster* plays a pivotal role in diversification of circular visceral muscle (*Schaub and Frasch, 2013*). This musculature is reduced in the Eriophyoidea (*Nuzzaci and Alberti, 1996*; *Whitmoyer et al., 1972*) compared to other mites (*Alberti and Crooker, 1985*; *Coons, 1978*; *Mathieson and Lehane, 2002*), as it also is in studied microinsects (*Polilov, 2015*). Furthermore, in most chelicerates, the Hox gene *pb* is expressed in the pedipalps and in three to four pairs of legs (*Barnett and Thomas, 2013*; *Schwager et al., 2015*; *Telford and Thomas, 1998*). Whether the lack of *pb* in the *A. lycopersici* genome is related to the reduction in legs in Eriophyoidea is unknown; however, *pb* has also been lost in other ecdysozoan

animals such as Nematoda (*Aboobaker and Blaxter, 2003*) and Tardigrada (*Smith et al., 2016*; *Yoshida et al., 2017*), lineages that either lack legs (Nematoda) or in which leg formation has been suggested to be highly aberrant ('walking heads', *Maderspacher, 2016*) from the panarthropodan ancestor (Tardigrada, *Smith and Goldstein, 2017*). Further, in *D. melanogaster* mutants of both *dachs* and *four-jointed*, each of which is absent in *A. lycopersici*, have similar phenotypes including shortened legs (*Buckles et al., 2001*). *A. lycopersici*-specific losses in cell cycle regulatory genes like *unkempt* and *fat* are also candidates to underlie allometric changes in tissues and organs, a general feature of diminutive mites (like *A. lycopersici*) and microinsects (*Danks, 2006*; *Polilov, 2015*).

A remarkable feature of the genome of *T. urticae* is the presence of hundreds of genes acquired from fungal or bacterial sources, including microbe-derived UGTs (*Wybouw et al., 2018*). While a modest number of UGTs of putative bacterial origin are present in *A. lycopersici*, horizontally transferred genes were otherwise absent, except for two genes in the pathway for the synthesis of pantothenate, an essential B vitamin. Previous studies have shown that pantothenate biosynthetic genes have been laterally transferred into tetranychid mites, the silverleaf whitefly, and nematodes (*Chen et al., 2016*; *Craig et al., 2009*; *Wybouw et al., 2018*)(*Ren et al., 2020*). In *A. lycopersici,* the HGT event of ketopantoate hydroxymethyltransferase appears to be distinct from the transfer in the tetranychid mite lineage. The apparent independent HGT of pantothenate biosynthetic genes in Acariformes, coupled with acquisitions in insect lineages, is a strong signal of adaptive significance for de novo pantothenate biosynthesis in arthropod herbivores.

Finally, nowhere were reductions in *A. lycopersici* gene families more striking than in genes associated with host plant use. Recently, the importance of chemosensory receptors in host plant use and breadth has attracted intense interest (*Gloss et al., 2019*; *Ngoc et al., 2016*). *A. lycopersici* completely lacks the expansion of chemosensory receptors reported (to varying extents) in nearly all other arthropods, as only a handful of members are present for any of the characterized chemosensory receptor families. This finding is consistent with a reduced role for chemosensation in specialist herbivores, although it may also reflect a more general loss of sensory structures during miniaturization, as the number of sensilla (which include sites of chemosensation) are dramatically reduced in microinsects (*Polilov, 2015*), as well as in eriophyoid mites (*Figure 1a,b*; *Lindquist and Oldfield, 1996*). Next to chemosensory receptor genes, the detoxification gene complement of *A. lycopersici* is minimal compared to the generalist herbivore *T. urticae* (*Dermauw et al., 2013b*; *Grbić et al., 2011*), as well as to insect herbivores (*Rane et al., 2019*). This was particularly striking for CCEs and GSTs, for which lineage-specific expansions are absent, and for which most members are in highly conserved clades that likely perform more general (non-detoxification) roles. Several of the few notable lineage-specific expansions in *A. lycopersici* do involve subfamilies of the MFS. However, while some MFS genes are differentially regulated upon host shift or xenobiotic exposure in *T. urticae* (*Dermauw et al., 2013b*), MFS proteins have diverse roles, and additional work is needed to assess if MFS mini-expansions in *A. lycopersci* are associated with host use.

The minimal detoxification gene repertoire and the paucity of chemoreceptor genes in *A. lycopersici* are in line with ecological specialization theory that predicts that herbivores with a narrow host range only need a limited number of environmental response genes (*Berenbaum, 2002*; *Rane et al., 2019*). However, although *A. lycopersici* has a narrow host-range relative to the spider mite *T. urticae*, it can be found on related solanaceous plant species (*Perring and Farrar, 1986*), as well as on several hosts outside the nightshade family (*Perring and Royalty, 1996*; *Rice and Strong, 1962*). Hence, the extent to which this mite has specialized on these hosts is unclear. Nevertheless, the minimal detoxification and chemoreception repertoire gene sets support the idea that modification of the local environment by defense suppression may alter selection imposed by the environment, thereby reducing the requirement for environmental response genes (*Laland et al., 2016*). How eriophyoids manipulate their hosts is unknown, but likely involves orally delivered salivary metabolites (*De Lillo and Monfreda, 2004*), or alternatively secreted proteins, termed effectors. Currently, the molecular nature of herbivore effectors, and their mechanisms of action, are poorly understood (*Blaazer et al., 2018*; *Erb and Reymond, 2019*). However, proteins secreted by the larvae of several lepidopteran species have been shown to attenuate plant defenses, including by physical interaction with a component of the JA signal transduction pathway (*Chen et al., 2019*; *Musser et al., 2002*). Further, a salivary ferritin from the whitefly *Bemisia tabaci* suppresses oxidative signals in tomato, and blunts JA-mediate defense responses (*Su et al., 2019*), and expression of salivary products of unknown molecular function from spider mites in plants was recently demonstrated

to impair defense signaling downstream of the phytohormone salicylic acid (*Villarroel et al., 2016*), and may also act to suppress JA signaling (*Schimmel et al., 2017*). The divergent molecular nature of these effectors mirrors findings from plant-pathogen (*Toruño et al., 2016*) and plant-nematode (*Rehman et al., 2016*) systems, where secreted effectors can be highly species-specific, hindering identification based solely on sequence information. These findings highlight the need for functional studies to establish if secreted proteins (or metabolites) in *A. lycopersici* saliva underlie this mite's ability to potently suppress tomato defenses. More generally, as additional genomes of herbivores that induce or suppress plant defenses become available – and that vary in their magnitude and mechanisms of host suppression – the *A. lycopersici* genome will serve as a key reference for comparative studies to test hypotheses surrounding the evolution of gene families that respond to or modulate plant defenses.

## Conclusion

At only 32.5 Mb, the *A. lycopersici* genome is the smallest sequenced arthropod genome to date. In contrast to its closest sequenced relatives, the majority of genes lack introns, few repetitive sequences are present, and many genes conserved in most animals are absent. Compared to its larger relatives, the simplification of *A. lycopersici*'s body plan, and that of eriophyoid mites more generally, is reminiscent of that observed in other microarthropods (*Maderspacher, 2016*). The compressed genome architecture of *A. lycopersici* is in line with genome streamlining concepts (*Hessen et al., 2010a*; *Hessen et al., 2010b*), some of which speculate that maintaining a high growth rate in nutritionally limited environments (in this study the plant epidermis) may be a driver for the evolution of compact genomes. Further, the extreme reduction of several environmental response gene families aligns with predictions that follow from ecological specialization theories (*Devictor et al., 2010*; *Futuyma and Moreno, 1988*; *Laland et al., 2016*) since the mite's suppression of plant defenses may allow for such families to minimize during the course of its evolution. Finally, this first eriophyoid genome provides a resource for methods of early detection of mite infestations using molecular markers, and its reduced complement of defense genes – a common source of pesticide resistance – may also reveal novel Achilles' heels for the control of *A. lycopersici*. But foremost, this genome is a milestone for accelerating our understanding of the evolutionary forces underpinning metazoan life at the limits of small physical and genome size.

## Materials and methods

### Collection of DNA for genomic sequencing

*A. lycopersici* individuals were reared in insect cages (BugDorm-44590DH, Bug Dorm Store, Mega-View Science, Taichung, Taiwan) in a walk-in growth chamber on tomato plants (*Solanum lycopersicum*, cv. Castlemart) that were between 3 and 6 weeks old. The climate room was set to day/night temperatures of 27°C/25°C, a 16/8 hr light/dark regime and 60% relative humidity. Harvesting of *A. lycopersici* mites was performed by detaching highly infested tomato leaflets and placing them in 1.5 mL Eppendorf tubes. Eppendorf tubes were filled with water and mites (adults, juveniles and eggs) were washed off by rinsing and briefly vortexing the tubes. The tubes were then centrifuged (13,000 rpm for 2 min), after which bulk tomato tissue was removed and water was pipetted away. Contamination from tomato tissue was limited to small amounts (less than ~5%) of material consisting primarily of tomato trichomes. Resulting 'pellets' of russet mites were frozen in liquid nitrogen and stored at −80°C until DNA was extracted.

### DNA sequencing and genome assembly

DNA was extracted using a modified version of the CTAB method (*Doyle and Doyle, 1987*). Sixty μg of DNA dissolved in TE buffer was sent to Eurofins MWG Operon (Ebersberg, Germany) for sequencing. Sequencing reads were produced with the standard Roche/454 sequencing protocol on the GS FLX system running Data Analysis Software Modules version 2.3. Three different libraries were prepared and sequenced in accordance with the recommendations of Roche/454: random primed shotgun, 8 kb paired-end, and 20 kb paired-end. From the shotgun library the mean length was 503 bp, while for the 8 kb and 20 kb libraries the mean lengths were 366 bp and 359 bp, respectively. Sequencing reads were trimmed to remove adapters and low-quality bases, as well as

to split each paired-end read into a forward and reverse pair; this yielded a total of 1,854,028 shotgun reads, 1,076,303 reads from the 8 kb library, and 1,274,414 reads from the 20 kb library. Contigs were assembled by the in-house pipeline of Eurofins MWG Operon (Ebersberg, Germany) based on Newbler 2.6 (*Margulies et al., 2005*). Following scaffolding and filtering for plant (tomato), prokaryotic, and adaptor sequences, a reference for the nuclear genome was generated that consisted of seven scaffolds (scaffolds 1, 2, 3, 4, 5, 11, and 17) with a total length of 32.53 Mb (the Newbler 'peakDepth', or coverage, for the assembly was 38). An additional scaffold (scaffold 6) of length 13.5 kb consisted of the mitochondrial genome.

## Genome size and completeness estimations

A k-mer size estimate of the *A. lycopersici* genome was performed using the genomic 454 sequence reads and Jellyfish 2.2.6 (*Marçais and Kingsford, 2011*). Following the recommendations of T. Nishiyama (http://koke.asrc.kanazawa-u.ac.jp/HOWTO/kmer-genomesize.html), genome size was estimated by running Jellyfish (*Marçais and Kingsford, 2011*) with the following settings '-t 24 iC -s 20M' for all odd k-mer values from 17 to 31, with averaging of the results provided from the eight different estimates. Completeness of the genome was also assessed using CEGMA 2.5 (*Parra et al., 2007*) as well as BUSCO v3 (*Simão et al., 2015*), as well as with an alignment of the *A. lycopersici* Illumina-based transcriptome assembly to the genomic scaffolds (see below, and Results section).

## RNA collection, 454 cDNA sequencing, and transcriptome assembly

Mixed developmental stages (adults, juveniles, and eggs) were collected from tomato leaflets as was done for DNA preparation. Similar to DNA extraction, small amounts of tomato trichome contamination were evident, but at low levels. RNA was extracted using a Qiagen RNeasy kit (Qiagen, Hilden, Germany) according to the manufacturer's instructions. Forty-five µg of RNA was provided to Eurofins MWG Operon for library preparation according to standard Roche protocols. Following poly(A) selection and strand-specific cDNA library preparation, the library was analyzed on a Shimadzu MultiNA microchip electrophoresis system (Shimadzu, Kyoto, Japan) to verify that the gel size selection was in the range of 500–800 bp. A total of 1,370,892 sequencing reads were collected using a Roche GS FLX system employing the Titanium series chemistry. After trimming of cDNA reads to remove low quality reads and adapter sequences, the remaining 1,370,005 reads were assembled using MIRA (*Chevreux et al., 2004*).

## RNA collection, Illumina sequencing, and transcriptome assembly

RNA was extracted from eight *A. lycopersici* pools using the Qiagen RNeasy purification kit (Qiagen, Hilden, Germany) with the following adaptations: Step 3: 50 µl of RNEasy lysis buffer (RLT) + ß -mercaptoethanol were added to the mite pool in a 1.5 mL tube, followed by 1–2 min of cell lysis performed by twisting and turning a 1.5 mL-tube-pestle. Three hundred µl of RLT + -mercaptoethanol was then used to rinse the pestle; Step 11: RNA was eluted in 30 µl RNAse-free water and stored on ice. All samples were stored at −20˚C. Strand-specific paired-end RNA library preparation and sequencing were carried out by the Centro Nacional de Análisis Genómico (Barcelona, Spain) to yield a total of 86.6 million 101 bp read pairs.

To construct a transcriptome assembly from the Illumina RNA-seq reads, the reads were first aligned to the *A. lycopersici* reference genome sequence using STAR 2.5.2b (*Dobin et al., 2013*) with the following settings: twopassMode Basic, sjdbOverhang 100, and alignIntronMax 20000. Reads that did not align to the reference were subsequently aligned against the tomato genome release SL 2.50 (*Tomato Genome Consortium et al., 2012*) to filter out contamination from the host plant with the same settings used to align to the mite genome except for alignIntronMax, which remained unspecified. The reads that did not align to the tomato genome were pooled with the reads that aligned to the *A. lycopersici* genome and imported into CLC Genomics Workbench 9.0.1 (https://www.qiagenbioinformatics.com/), where they were trimmed using the default parameters (quality score limit 0.05 and a maximum of two ambiguous nucleotides) before being assembled with the default settings and a minimum contig length of 200. The resulting 13,428 transcript sequences were aligned back to the *A. lycopersici* genome assembly using BLAST 2.5.0+ (*Camacho et al., 2009*) to provide a measure of the genome completeness for transcribed regions. Of the 243 transcripts that did not align, 23 had no hits in any database, and 108, 84 and 20

appeared to be from bacterial, plant and fungal sources, respectively. Only eight had homology to arthropod sequences present in the NCBI NR, NT, Other Genomic, RefSeq Genomic, RefSeq RNA, Representative Genomes, and WGS databases (downloaded January 9, 2017).

## Annotation of the *Aculops lycopersici* genome

A first-pass annotation was produced using EuGene (*Schiex et al., 2001*) specifically trained for the studied genome using the 454 transcript read data as a guide. As a consequence of the close proximity of adjacent genes (see Results and *Table 1*), we observed that transcript contigs often merged adjacent genes, creating apparent chimeric genes. To circumvent this issue, only junctions spanning introns as assessed from the aligned 454 data were kept after mapping. Besides using transcript data, protein homology to the invertebrate section from RefSeq, curated proteins from SWISSprot and the proteome from *T. urticae* were used.

Subsequently, the annotation was revised in several ways. The deep dataset of Illumina RNA-seq reads was aligned to the genome using the default settings of Bowtie 2.2.3 (*Langmead and Salzberg, 2012*)/TopHat 2.0.12 (*Kim et al., 2013*), as well as STAR 2.5.2b (*Dobin et al., 2013*) with the parameters described previously. Transcripts from the CLC transcriptome assembly were also located on the genome using BLAT 36 (*Kent, 2002*). Then, Cufflinks 2.2.1 (*Trapnell et al., 2013*) and TransDecoder (Release 20140704) (*Haas et al., 2013*) were used to identify additional ORFs of over 300 bp in length that had not been detected by EuGene. Resulting gene models were then added where supported by the strand-specific RNA-seq reads and/or transcript alignments. The compact nature of the *A. lycopersici* genome, coupled with the finding that most genes were intronless (*Table 1*), made it feasible to then manually inspect all gene models against the aligned Illumina RNA-seq read data. This inspection step was performed using the Integrative Genomics Viewer (*Robinson et al., 2011*), which allowed simultaneous display of gene models and RNA-seq read alignments. Manual adjustments to gene models, where required, were performed using Genome-View N29 (*Abeel et al., 2012*). Additionally, members of specific gene families were expertly annotated as described in the section 'Comparative analyses with specific gene families', with resulting adjustments also incorporated in the final annotation. GenomeTools 1.5.10 (*Gremme et al., 2013*) was used to sort, correct phase information, and validate the resulting GFF3.

## Genome metric calculations

Coding gene numbers and the percentages of intronless genes were calculated with the 'stat -exon-numberdistri' command of the GenomeTools 1.5.6 package (*Gremme et al., 2013*) using the respective GFF3 annotation files as input (*Table 1*). Where multiple isoforms were present for a gene, only the longest isoform was used for this and subsequent analyses. Regions of the respective genomes were then classified as coding, intergenic or intronic by parsing the location of coding sequences (CDS) from the respective GFF3 annotation files; due to the unreliability of untranslated sequence prediction or their complete absence in some annotations, these regions were not considered. In instances where CDS sequences overlapped, their coordinates were merged so that no region of the genome would be counted multiple times. Regions of the genome between the start and end of the CDS sequences of adjacent genes were classified as intergenic, while regions of the genome within genes that did not fall into CDS coordinate blocks were classified as intronic (in instances where genes were located within the introns of other genes, the CDS sequences of the genes within the introns were classified as coding, with the remaining portion counted as intronic).

## Transposable element annotation

The consensus of the repeated DNA ($\geq$2 copies) in the genome was constructed by employing RepeatScout (v.1.0.5) (*Price et al., 2005*). The repeats that were $\geq$90% identical with a minimum overlap of 40 bp were assembled using CAP3 (*Huang and Madan, 1999*). Gene families were identified based on homology with cellular genes by employing tBLASTx 2.2.28+ (*Altschul et al., 1997*) searches against the Refseq mRNA database at NCBI and BLASTn 2.2.28+ (*Altschul et al., 1997*) searches against the annotated genes in the *A. lycopersici* genome. All candidate gene families were filtered upon manual verification. The remaining repeats were classified by REPCLASS (*Feschotte et al., 2009*) and RepeatMasker (*Smit et al., 2013*) protein searches (http://www.repeatmasker.org/cgi-bin/RepeatProteinMaskRequest). The repeats that were classified based on the

structure or TSD module of REPCLASS were manually verified. The criteria of requiring at least one defined end were used to classify a repeat as a TE. To identify if the elements had at least one defined end, the unclassified repeats (≥65 bp) were aligned with the respective copies with extended flanking sequences using MUSCLE (*Edgar, 2004*). Repeats were classified and full-length copies were extracted when possible. To identify low copy non-LTR retrotransposons, the non-LTR proteins from the related mite *T. urticae* were used as queries in homology-based tBLASTn 2.2.25+ (*Altschul et al., 1997*) searches against the *A. lycopersici* genome. To identify the genomic coverage, the curated repeat library was used to mask the genome using RepeatMasker (v 4.0.5) (*Smit et al., 2013*). The final RepeatMasker output was parsed using parseRM.pl (*Kapusta et al., 2017*; *Kapusta, 2017*) to identify the contribution of TEs (*Figure 2—figure supplement 1*, *Supplementary file 1* — 'Table S1' Tab). Last, a gene and TE density plot was constructed using karyoploteR version 1.14.0 (*Gel and Serra, 2017*) and the GFF3 annotation file of the *A. lycopersici* genome (*Table 1—source data 1*) and the RepeatMasker output (*Supplementary file 1* — 'Table S2' Tab), respectively.

## Analysis of intronic features

The longest protein isoforms for the following organisms were extracted for orthologue identification: *A. lycopersici* (current genome), *Anopheles gambiae* AgamP4.7 (*Holt et al., 2002*), *Bombyx mori* ASM15162 (Ensembl release 37) (*Mita et al., 2004*), *Caenorhabditis elegans* Wormbase release WS261 (The *C. elegans The C. elegans Sequencing Consortium, 1998*), *Centruroides sculpuratus* CEXE 0.5.3 (*Schwager et al., 2017*), *Danio rerio* GRCz10 (Ensembl release 89) (*Howe et al., 2013*), *Daphnia pulex* PA42 3.0 (*Ye et al., 2017*), *Dermatophagoides pteronyssinus* (ASM190122v2) (*Waldron et al., 2017*), *Drosophila melanogaster* Flybase release 6.16 (*Adams et al., 2000*; *Gramates et al., 2017*), *Homo sapiens* GRCh38.p10 (Ensembl release 89) (*Lander et al., 2001*; *Venter et al., 2001*), *Ixodes scapularis* (IscaW1.5) (*Gulia-Nuss et al., 2016*), *Limulus polyphemus* 2.1.2 (*Simpson et al., 2017*), *Metaseiulus occidentalis* 1.0 (GNOMON release) (*Hoy et al., 2016*), *Parasteatoda tepidariorum* 1.0 (*Schwager et al., 2017*), *Pediculus humanus* PhumU2 (Ensembl release 36) (*Kirkness et al., 2010*), *Strigamia maritima* Smar1 (Ensembl release 36) (*Chipman et al., 2014*), *T. urticae* (ORCAE August 11, 2016 release) (*Grbić et al., 2011*), and *Tribolium castaneum* Tcas5.2 (Ensembl release 36) (*Richards et al., 2008*). The identification of orthologous protein sequences was performed with OrthoFinder 1.1.8 (*Emms and Kelly, 2015*) using BLAST 2.6.0+.

We found 147 single-copy orthologues across all species that we then aligned using MAFFT 7.305b (*Katoh and Standley, 2013*) with 'genafpair' and 'maxiterate 1000'; a concatenation of the alignments for the 147 orthologues was then generated prior to trimming with trimAl 1.4.rev15 (*Capella-Gutiérrez et al., 2009*) using the 'strictplus' option. The trimmed sequences were used for a phylogenetic reconstruction with RAxML 8.2.12 (*Stamatakis, 2014*) using the LG+G+F model as identified for phylogenetic reconstruction by ProtTest 3.4.2 (*Darriba et al., 2011*) according to the Akaike Information Criterion, and a total of 1000 rapid bootstrap replicates ('-f a -x 12345' option). Although the 'estimate proportion of invariable sites (+I)' was also recommended by ProtTest, the developer of RAxML v8, on page 59 of the RAxML v8.2.X manual, cautions against using this option, and this and all subsequent optimal models for reconstructions with RAxML were adjusted to adhere to this developer recommendation.

Orthologous protein clusters were selected for intron analysis on the basis of the following criteria: the cluster had to have at least one orthologue from *A. lycopersici*, orthologous protein sequences from at least 14 other species had to be present, and no species could have more than three orthologous proteins in the cluster; when multiple orthologues for a single species were present, only the longest one was retained. The sequences in these clusters were aligned using MAFFT 7.305b (*Katoh and Standley, 2013*) with the settings previously described. GNU Parallel (*Tang, 2011*) was used to align multiple clusters at once. Custom Python scripts using BioPython 1.70 (*Chapman and Chang, 2000*) and the BCBio GFF parser (*Chapman, 2016*) were used to parse and append intron position information to the FASTA sequence identifier line as required by Malin (*Csurös, 2008*). The 2371 clusters that met the requisite criteria, along with the tree built from the 147 single-copy orthologues, were used in the Malin analysis (*Csurös, 2008*). Intron positions for gain/loss analysis were selected from those that were considered unambiguous in *A. lycopersici* and at least 11 other species, with five amino acids present on either side of the intron position (a Malin criteria to reduce the possibility of incorrect inference resulting from misalignments).

To investigate the consequence of intron losses in *A. lycopersici* on predicted protein sequences, which can shed light on underlying mechanisms of loss (see Discussion), a subset of orthogroups was selected for which sequences for each of *A. lycopersici*, *D. pteronyssinus*, *T. urticae*, *M. occidentalis*, *B. mori* and *D. melanogaster* were present as single copies (1216 in total); apart from *A. lycopersici*, the five other species were selected because of their close phylogenetic position to *A. lycopersici* (*Figure 3*), and/or because they have high-quality genomes and annotations. The protein sequences for the six species for each of these orthogroups were aligned using MAFFT 7.407 with the settings previously described, and a table of intron sites for these orthogroups was generated in Malin using the following settings: Minimum nongap positions: 0 (On both sides); Minimum unambiguous characters at a site: 1; There must be at least one unambiguous character in the following clades: All unselected. From this table, intron positions that were present and had the same phase in all arthropods except *A. lycopersici* (indicating a high degree of conservation), and for which Malin identified a missing intron in a region of unambiguous alignment for *A. lycopersici* sequences, were manually examined across all protein sequence alignments to assess if intron loss events in the respective genes introduced gains or losses of residues in the encoded products. The classification results for these sites (100 in total among 80 orthogroups), are included in *Supplementary file 1* — 'Table S3' Tab; the sequence alignments and annotations of intron positions for each orthogroup are given in *Supplementary file 3*.

## Gene family expansions and contractions

The OrthoFinder analysis (see section 'Analysis of intronic features') generated 86,686 orthologous groups (OGs) in total, of which 13,817 contained more than one protein (*Supplementary file 1* — 'Table S7' Tab). InterProScan 5.25–64.0 (*Quevillon et al., 2005*) was run to assign domains to each of the proteins in all 18 species, and the domain information was subsequently assigned to the OrthoFinder OGs using the KinFin software (*Laetsch and Blaxter, 2017*) and an associated Python script (functional_annotation_of_clusters.py with the options: '–p 0.3 and –x 0.3'). Two different strategies were used to identify contracted and/or expanded gene families in *A. lycopersici*. First, we used the CAFE software to detect contracted/expanded orthologous groups (orthogroups, OGs) among 18 metazoan species, while the second strategy was focused on OG expansions within the acariform mites, *A. lycopersici*, *D. pteronyssinus* and *T. urticae* using an arbitrary rule. OrthoFinder 1.1.8 (*Emms and Kelly, 2015*) with BLAST 2.6.0+ was used to identify OGs among the proteomes of 18 metazoan species (see 'Analysis of intronic features' for proteome versions that were used as input for OrthoFinder).

To maximize the probability of achieving convergence in the maximum likelihood analysis performed in CAFE, OGs were processed to remove OGs present in only a few species and were subsequently divided into OGs having <100 gene copies in any species ('small' OGs) and orthogroups having one or more species with ≥100 gene copies ('large' OGs); see 'Known Limitations' section in CAFE 4.0 Manual of March 14, 2017 and section 2.2.4 of the CAFE 4.0 tutorial online at https://iu. app.box.com/v/cafetutorial-pdf, and also *Casola and Koralewski, 2018*. We retained 6,496 OGs that occurred in no less than 10 out of 18 species consisting of 6,467 'small' OGs and 29 'large' OGs. Together with an ultrametric species tree the 'small' OG dataset was used as input in CAFE to estimate the birth/death parameter λ (the probability that a gene will be gained or lost) and to identify rapidly evolving OGs (p-value threshold of 0.05). The estimated λ (0.00055594301461) was then used to identify rapidly evolving OGs in a CAFE analysis with 'large' OGs and using the same p-value threshold and ultrametric species tree as in the CAFE analysis with 'small' OGs. The ultrametric species tree used as input in both CAFE analyses was obtained by using the species tree generated for the Malin intron analysis, subsequently rooting this tree using vertebrates as outgroup, and converting this rooted tree into an ultrametric tree using the *convert_to_ultrametric()* command in the *Tree* package of the ETE toolkit (ete 3.0.0b35) (*Huerta-Cepas et al., 2016*). Next, branch lengths of the ultrametric tree were scaled to time units using the software treePL (*Smith and O'Meara, 2012*) with the following options: 'smooth = 0.01, numsites = 41107 (number of sites in the alignment used for the Malin analysis), thorough, opt = 4, moredetailad, optad = 2, optcvad = 2, moredetailcvad' and using seven calibration timepoints: the divergence time between Eriophyoidea and Sarcoptiformes (352–410 MYA), Sarcoptiformes and Trombidiformes (410–421 MYA) and Mesostigmata and Ixodida (283–418 MYA) as derived from *Xue et al., 2017*, and the divergence time between *D. melanogaster* and *A. gambiae* (211–335 MYA), Scorpiones and Araneae (379–410 MYA), Mandibulata

and Chelicerata (560–642 MYA) and *H. sapiens* and *D. rerio* (425–446 MYA), as obtained from Time-Tree (*Kumar et al., 2017*) on February 20, 2019. The options used in treePL were determined following the 'Quick run' guidelines of the treePL wiki (*Smith, 2012*). The output of the two CAFE analyses ('small' and 'large' OGs) was summarized using a Python script (cafetutorial_report_analysis.py using the '-l' option and with a *p*-value cutoff set to 0.05) available at the CAFE tutorial website (https://iu.app.box.com/v/cafetutorial-files/folder/22161236877, accessed February 20, 2019). The tree with OG expansions and contractions was visualized in MEGA 6.0 (*Tamura et al., 2013*) and edited with Corel Draw software (Corel Draw, Inc); the list of rapidly evolving (expanding or contracting) OGs can be found in *Supplementary file 1* — 'Table S5' Tab. Rapidly contracting *A. lycopersici* gene families with more than ten members were analyzed for the percentage of *A. lycopersici* members showing orthology with the majority of chelicerate species (*Supplementary file 1* — 'Table S6' Tab). Orthology was determined based on the Orthofinder generated output in the 'Orthologues_Aculops_lycopersici' folder. One of the six rapidly contracted *A. lycopersici* families belonged to the CYP family and was excluded from the analysis, as only few orthology relationships has been observed within this family (*Feyereisen, 2011*).

Apart from gene families that we identified as expanded in the high-level CAFE analysis, we looked as well for more subtle expansions across acariform mites. Across all orthogroups identified by Orthofinder, we identified eleven orthogroups with (1) *A. lycopersici* having more than five members and (2) *A. lycopersici* having twofold more members than the average number in *T. urticae* and *D. pteronyssinus* (OG0000024, OG0000271, OG0000546, OG0000706, OG0004829, OG0006109, OG0006384, OG0007553, OG0007554, OG0008410, OG0008412). For two orthogroups (OG0007554, OG00084112), no InterPro domain could be assigned, while OG0000271, OG0000706, OG0004829, OG0006384, OG0007553, and OG0008410 contained proteins with a DnaJ domain (IPR011701), Formate-tetrahydrofolate ligase domain (IPR000559), Acyltransferase 3 domain (IPR002656), a Peptidase C1A domain (IPR000668), Chromo domain (IPR023780) and a Lipase/vitellogenin domain (IPR013818), respectively. The proteins of the three remaining orthogroups (OG0000024, OG0000546, and OG0006109) belonged to the Major facilitator superfamily (MFS, IPR011701 or IPR024989).

## Gene ontology enrichment analysis of absent conserved genes and identification of orthogroups containing *Drosophila* essential genes

For *D. melanogaster* proteins belonging to orthogroups with (1) members in all included arthropods except *A. lycopersici* and (2) a maximum of two *D. melanogaster* members (343 *D. melanogaster* proteins in total, *Supplementary file 1* — 'Table S8' Tab), we performed an Over-Representation analysis (ORA) using the WEB-based GEne SeT AnaLysis Toolkit (*Liao et al., 2019*). An ORA was performed for each GO category (Biological Process, Molecular Function and Cellular Component) using default settings (and 'genome protein coding' as reference set) and a Benjamini-Hochberg multiple testing correction (false discovery rate, FDR, of 0.05). In addition, we also identified those orthogroups that contain purported *D. melanogaster* essential genes, using the list of 427 essential genes provided in the respective study's first supplementary data table (*Aromolaran et al., 2020*).

## Comparative analyses with specific gene families

We specifically analyzed genes and gene families associated with herbivory in other animals, as well as those associated with physiological or developmental process related to *A. lycopersici*'s life history or derived morphology (GSTs, CCEs, CYPs, ABC transporters, MFS proteins, proteases, chemosensory receptors, and transcription factors, including Hox genes). We also characterized genes involved in processes including circadian rhythm, small RNA pathways, and potential regulation of plant defense responses (secreted proteins).

### Characterization of detoxification and feeding associated gene families
#### Glutathione-S-transferases

The *A. lycopersici* genome and proteome were mined for glutathione-S-transferases (GSTs) by tBLASTn and BLASTp searches, respectively, using cytosolic and microsomal *T. urticae* GST protein sequences as query (*Grbić et al., 2011*) and an E-value threshold of $E^{-5}$. In total, four *A. lycopersici* cytosolic GSTs were identified. *A. lycopersici* cytosolic GSTs were aligned with those of *T. urticae* (31

GSTs) (*Grbić et al., 2011*), *D. melanogaster* (36 GSTs, the atypical GST CG4623/Gdap1 was not included as it is very divergent from other *D. melanogaster* GSTs) (*Shi et al., 2012*) and *M. occidentalis* (13 GSTs) (*Wu and Hoy, 2016*) using the online version of MAFFT v7.356b (*Katoh and Standley, 2013*) with 1000 iterations with the options 'E-INS-i' and 'reorder' (see *Supplementary file 7*). Model selection was performed with ProtTest 3.4 (*Darriba et al., 2011*), and according to the Akaike information criterion LG+I+G+F was optimal for the phylogenetic reconstruction. A maximum likelihood analysis was performed using RAxML v8 HPC2-XSEDE (*Stamatakis, 2014*) on the CIPRES Science Gateway (*Miller et al., 2010*) with 1000 rapid bootstrapping replicates ('-f a -x 12345' option). The resulting tree was midpoint rooted, visualized using MEGA 6.0 (*Tamura et al., 2013*) and edited with Corel Draw software (Corel Draw Inc).

## Carboxyl/cholinesterases

Putative carboxyl/cholinesterase (CCE) genes were identified in *A. lycopersici* using tBLASTn and BLASTp searches (E-value threshold of $E^{-5}$) with *T. urticae* CCE sequences (*Grbić et al., 2011*) as query. Putative *A. lycopersici* CCEs were aligned with those of *T. urticae* (*Grbić et al., 2011*), *M. occidentalis* (*Wu and Hoy, 2016*), a selection (8) of conserved CCEs from the horseshoe crab *Limulus polyphemus* (*Wei et al., 2020*), a selection (10) of *D. melanogaster* CCEs belonging to different CCE clades (*Claudianos et al., 2006*), and AChE1/AChE2 of *B. mori* and *D. pulex* using the online version of MAFFT v7.380 (*Katoh et al., 2019*) with 1000 iterations and the options 'L-INS-i' and 'reorder' (see *Supplementary file 7*). Maximum likelihood phylogenetic analysis was performed as in *Wei et al., 2020* using RAxML v8 HPC2-XSEDE (*Stamatakis, 2014*) on the CIPRES Science Gateway (*Miller et al., 2010*) and the automatic protein model assignment algorithm using maximum likelihood criterion and 500 rapid bootstrap replicates ('-f a -x 12345' option). The resulting tree was midpoint rooted and visualized using MEGA 6.0 (*Tamura et al., 2013*) and edited with Adobe Illustrator software (Adobe Inc).

## Cytochrome P450 monooxygenases and diflavin reductases

The *A. lycopersici* genome and proteome was mined for cytochrome P450 monooxygenase (CYP) genes by tBLASTn and BLASTp searches using *T. urticae* CYP protein sequences as query (*Grbić et al., 2011*) and an E-value threshold of $E^{-5}$. All CYP gene models with predicted proteins that included the canonical heme-binding sequence were verified manually for the presence of the other key features of P450 enzymes (*Feyereisen, 2012*) and gene models were corrected when necessary. New *A. lycopersici* CYP gene models were created using GenomeView (*Abeel et al., 2012*). All CYP sequences were named according to the CYP nomenclature by Dr. D. R. Nelson (University of Tennessee, USA). Pseudogenes (*CYP18C2P* and *CYP3120A4P*) were distinguished from putative full length CYP coding sequences by a long in frame non-P450 insertion (*CYP18C2P*) and by a stop codon and frameshift (*CYP3120A4P*), both anomalies confirmed by their respective transcripts. All *A. lycopersici* CYP protein sequences (full-length and pseudogenes) were aligned with CYP protein sequences from *T. urticae, M. occidentalis*, a set of *D. melanogaster* marker P450 sequences and the CYP18 protein sequence of the house dust mite *D. pteronyssinus* (Dpte.g6170.1) using MAFFT v7.380 (*Katoh et al., 2019*) with 1000 iterations and the options 'E-INS-i' and 'reorder' (see *Supplementary file 7*). Model selection was done with ProtTest 3.4 (*Darriba et al., 2011*) and according to the Akaike information criterion LG+I+G+F was optimal for phylogenetic reconstruction. A maximum likelihood analysis was performed using RAxML v8 HPC2-XSEDE (*Stamatakis, 2014*) on the CIPRES Science Gateway (*Miller et al., 2010*) with 1000 rapid bootstrapping replicates ('-f a -x 12345' option). The resulting tree was midpoint rooted and visualized using MEGA 6.0 (*Tamura et al., 2013*).

## ABC transporters

Putative *A. lycopersici* ATP-binding cassette (ABC) genes were identified by BLASTp and tBLASTn searches (E-value threshold of $E^{-5}$) against the *A. lycopersici* proteome and genome, respectively, and using *T. urticae* ABC protein sequences (*Dermauw et al., 2013a*) as query. *A. lycopersici* ABC pseudogenes and incomplete genes [aculy01g37790, aculy01g37820 (pseudogenes), and aculy01g27210 (incomplete gene)] were separated from putative full-length ABC coding sequences. Putative *M. occidentalis* ABC genes were identified by a BLASTp search against the *M. occidentalis* proteome using *T. urticae* and *D. melanogaster* ABC protein sequences as query (*Dermauw et al.,*

2013a). The nucleotide-binding domain (NBD) sequences of *A. lycopersici, T. urticae, M. occidentalis*, and *D. melanogaster* ABC protein sequences were extracted using the ScanProsite facility (*de Castro et al., 2006*) and the Prosite profile PS50893. The NBDs of four putative *M. occidentalis* ABC proteins (GNOMON-2147495233, GNOMON-2147494257, GNOMON-2147494305 and GNO-MON-2147512403) had best BLASTp hits with bacterial ABC sequences and were excluded from further analysis. N-terminal NBDs of *A. lycopersici* (44), *T. urticae* (103), *M. occidentalis* (55), and *D. melanogaster* (56) ABC proteins were aligned using the online version of MAFFT v7.380 (*Katoh et al., 2019*) with 1000 iterations and the options 'G-INS-i' and 'reorder' (see *Supplementary file 7*). Model selection was performed with ProtTest 3.4 (*Darriba et al., 2011*) and according to the Akaike information criterion LG+G+F was optimal for phylogenetic reconstruction. Next, a maximum likelihood analysis was performed using RAxML v8 HPC2-XSEDE (*Stamatakis, 2014*) on the CIPRES Science Gateway (*Miller et al., 2010*) with 1000 rapid bootstrapping replicates ('-f a -x 12345' option). The resulting tree was midpoint rooted and visualized using MEGA 6.0 (*Tamura et al., 2013*) and edited with Adobe Illustrator software (Adobe Inc).

## Major facilitator superfamily proteins

*A. lycopersici* members of two orthogroups (OG0000024 and OG0006109) that have an MFS Inter-Pro domain (IPR011701 or IPR024989), and were expanded in *A. lycopersici* (see Results), were used as query in tBLASTn and BLASTp searches (with an E-value threshold of $E^{-5}$) against the *A. lycopersici* genome and proteome, respectively. Next, the *A. lycopersici* queries and resulting hits were used as query in tBLASTn and BLASTp searches (with an E-value threshold of $E^{-5}$) against the genome and proteome of *T. urticae* and in a BLASTp search (using an E-value threshold of $E^{-5}$) against the proteomes of *M. occidentalis* and *D. melanogaster* (for genes with multiple isoforms, only the longest protein isoform was retained). Incomplete MFS genes (less than 250 amino acids long) were separated from full-length MFS genes. Full-length MFS proteins (*A. lycopersici*: 27, *T. urticae*: 23, *M. occidentalis*: 60 and *D. melanogaster*: 18) were aligned using MAFFT v7.356b (*Katoh and Standley, 2013*) with 1000 iterations with the options 'E-INS-i' and 'reorder' (see *Supplementary file 7*). Model selection was done with ProtTest 3.4 (*Darriba et al., 2011*) and according to the Akaike information criterion LG+I+G+F was optimal for the phylogenetic reconstruction of mite and *D. melanogaster* MFS proteins. A maximum likelihood analysis was performed using RAxML v8 HPC2-XSEDE (*Stamatakis, 2014*) on the CIPRES Science Gateway (*Miller et al., 2010*) with 1000 rapid bootstrapping replicates ('-f a -x 12345' option). The resulting tree was midpoint rooted, visualized using MEGA 6.0 (*Tamura et al., 2013*) and edited with Corel Draw software (Corel Draw, Inc).

## C1A cysteine proteases

OG0006384, one of the few expanded OGs in *A. lycopersici* contained proteins with a 'Peptidase C1A, papain C-terminal' domain (InterPro domain IPR000668) (see Results). Subsequently the complete proteome of *A. lycopersici, M. occidentalis* and *D. melanogaster* was mined for IPR000668 domain containing proteins/C1A peptidases. *T. urticae* C1A peptidases were previously annotated (*Grbić et al., 2011*). Thirty-nine, 16, 28, and 57 C1A peptidase genes were found in *A. lycopersici, M. occidentalis, D. melanogaster*, and *T. urticae*, respectively. Protein sequences from 32, 13, 27, and 52 *A. lycopersici, M. occidentalis, D. melanogaster* and *T. urticae* C1A peptidase genes were larger than 250 aa, respectively, and aligned using MAFFT version 7 (*Katoh and Standley, 2013*) with 1000 iterations with the options 'E-INS-i' and 'reorder' (see *Supplementary file 7*). Model selection was performed with ProtTest 3.4 (*Darriba et al., 2011*), and according to the Akaike information criterion VT+G was optimal for phylogenetic reconstruction. A maximum likelihood analysis was performed using RAxML v8 HPC2-XSEDE (*Stamatakis, 2014*) on the CIPRES Science Gateway (*Miller et al., 2010*) with 1000 rapid bootstrapping replicates ('-f a -x 12345' option) and the VT+G model. The resulting tree was midpoint rooted, visualized using MEGA 6.0 (*Tamura et al., 2013*) and edited with Corel Draw software (Corel Draw Inc).

## Gustatory receptors

Potential gustatory receptor (GR) genes were identified with BLASTp using query gustatory receptor sequences from *D. melanogaster* (*Robertson et al., 2003*), *D. pulex* (*Peñalva-Arana et al., 2009*), *M. occidentalis* (*Hoy et al., 2016*), and *T. urticae* (*Ngoc et al., 2016*), as well as odorant receptor

sequences from *D. melanogaster* (*Robertson et al., 2003*). Further, searches were performed with query sequences against the *A. lycopersici* genome using tBLASTn from the BLAST 2.6.0+ (*Camacho et al., 2009*) suite allowing an E-value of up to 1 (*Ngoc et al., 2016*). Where required, existing gene models were modified or new models were added using GenomeView N29 (*Abeel et al., 2012*). InterProScan 5.25–64.0 (*Quevillon et al., 2005*) was used to validate one of the two existing genes (*aculy03g00430*) with the '7tm Chemosensory Receptor' InterPro domain, while strong BLAST support against *T. urticae* GR genes (best hit E-value $<E^{-22}$) was observed for *aculy03g06080*. The two putative GR genes were then aligned back to the genome with tBLASTn to identify additional sequences, but none were identified. Protein sequences of GR genes from *A. lycopersici*, *T. urticae*, *M. occidentalis*, and *D. melanogaster,* were aligned using version 7 of MAFFT (*Katoh and Standley, 2013*) with the 'E-INS-i' option selected on the web service hosted by the Computational Biology Research Consortium (https://mafft.cbrc.jp/alignment/server/) (see *Supplementary file 7*). Model selection was performed by ProtTest 3.4.2 (*Darriba et al., 2011*), with the JTT+I+G+F model selected as the best according to the Akaike information criterion for phylogenetic reconstruction. The CIPRES Science Gateway (*Miller et al., 2010*) 'RAxML-HPC on XSEDE' tool (*Stamatakis, 2014*) was used to construct a phylogenetic tree using 1000 rapid bootstrap replicates ('-f a -x 12345' option), which was subsequently visualized using MEGA7 (*Kumar et al., 2016*, p. 7) and edited in Adobe Illustrator CC 2017 (Adobe Software, Inc).

## Degenerin/epithelian Na+ channels

Candidate *A. lycopersici* degenerin/epithelial Na+ Channels (ENaC) genes were identified by aligning *D. melanogaster* (*Zelle et al., 2013*) and *T. urticae* ENaCs (*Ngoc et al., 2016*) against the *A. lycopersici* genome using tBLASTn, allowing an E-value of up to 1. Gene model adjustment or creation was performed with GenomeView N29. The presence of the Pfam PF00858 domain ('Amiloride-sensitive sodium channel') identified using InterProScan 5.25–64 was used as an additional criteria to identify ENaCs genes, see *Ngoc et al., 2016*. An additional round of genomic searches using tBLASTn with the four *A. lycopersici* ENaCs revealed no additional members. *A. lycopersici*, *T. urticae*, *M. occidentalis* and *D. melanogaster* ENaC protein sequences were aligned using MAFFT version 7 (*Katoh and Standley, 2013*) with the 'E-INS-i' option selected on the web service hosted by the Computational Biology Research Consortium (https://mafft.cbrc.jp/alignment/server/) (see *Supplementary file 7*). WAG+I+G+F was identified as the best model for phylogenetic reconstruction according to the Akaike information criterion by ProtTest 3.4.2 (*Darriba et al., 2011*). The 'RAxML-HPC on XSEDE' tool (*Stamatakis, 2014*) hosted by the CIPRES Science Gateway (*Miller et al., 2010*) was used to construct a phylogenetic tree with 1000 rapid bootstrap replicates ('-f a -x 12345' option). MEGA7 (*Kumar et al., 2016*) was used to visualize the resulting tree, which was subsequently edited in Adobe Illustrator CC 2017 (Adobe Software, Inc).

## Ionotropic receptors

Putative ionotropic receptor (IR) and related genes were identified by aligning *D. melanogaster* (*Gramates et al., 2017*) and *T. urticae* IRs (*Ngoc et al., 2016*) along with ionotropic glutamate receptors (iGluR) and glutamate ionotropic receptor NMDA type (GRIN) sequences from *T. urticae* (*Ngoc et al., 2016*) to the *A. lycopersici* reference genome using tBLASTn with an E-value of up to one allowed. Where appropriate, GenomeView N29 was used to manually adjust or create new gene models based on the alignments. BLAST hits with E-values $<E^{-5}$, in combination with detection of the Pfam domains PF00060, PF01094 and/or PF10613 were used to classify *A. lycopersici* genes as members of the IR/iGluR/GRIN group (*Ngoc et al., 2016*). Iterative tBLASTn searches with the 10 identified sequences to the *A. lycopersici* genome identified no additional candidates. IR/iGluR/GRIN protein sequences from *A. lycopersici*, *T. urticae*, *M. occidentalis*, and *D. melanogaster* were aligned using the Computational Biology Research Consortium's MAFFT version 7 (*Katoh and Standley, 2013*) web service (https://mafft.cbrc.jp/alignment/server/) with the 'E-INS-i' option selected excepting *M. occidentalis* non-IR sequences, which were not provided in the supplementary files of *Hoy et al., 2016* (see *Supplementary file 7*). ProtTest 3.4.2 (*Darriba et al., 2011*) identified LG+I+G+F as the best model for phylogenetic construction according to the Akaike information criterion. A phylogenetic tree was constructed using the 'RAxML-HPC on XSEDE' tool (*Stamatakis, 2014*) hosted by the CIPRES Science Gateway (*Miller et al., 2010*) with 1000 rapid bootstrap replicates ('-f a -x 12345' option). Visualization of the tree was performed using MEGA7

(*Kumar et al., 2016*, p. 7), with further edits carried out in Adobe Illustrator CC 2017 (Adobe Software, Inc).

## Transient receptor potential channels

The transient receptor potential (TRP) channel sequences for *D. melanogaster*, *M. musculus*, *M. occidentalis*, and *T. urticae* identified by *Peng et al., 2015* were downloaded from Ensembl (for *D. melanogaster*, *M. musculus*, and *M. occidentalis*) or ORCAE (*T. urticae*) using the IDs provided in that study. These protein sequences were aligned to the *A. lycopersici* genome sequence using tBLASTn 2.6.0+ to identify candidate TRP genes using an E-value of $E^{-10}$, as in *Peng et al., 2015*. Where appropriate, gene models were manually updated with GenomeView N29 using a combination of the BLAST alignments and transcriptome data. TRP channel protein sequences for *A. lycopersici*, *T. urticae*, *M. occidentalis*, and *D. melanogaster* were aligned using MAFFT version 7 (*Katoh and Standley, 2013*) with the 'E-INS-i' option selected using the web service hosted by the Computational Biology Research Consortium (https://mafft.cbrc.jp/alignment/server/) (see *Supplementary file 7*). The LG+I+G+F model was identified as optimal for phylogenetic construction by ProtTest 3.4.2 (*Darriba et al., 2011*) according to the Akaike information criterion. A phylogenetic tree was generated using the 'RAxML-HPC on XSEDE' tool (*Stamatakis, 2014*) hosted on the CIPRES Science Gateway (*Miller et al., 2010*), with 1000 rapid bootstrap replicates ('-f a -x 12345' option) and members of the Shaker family set as an outgroup after *Peng et al., 2015* to anchor the tree. MEGA7 (*Kumar et al., 2016*) was used to visualize the tree, with subsequent editing performed in Adobe Illustrator CC 2017 (Adobe Software, Inc).

## Characterization of transcription factors

Pfam domains that were assigned by InterProScan to each of the proteins in the 18 metazoan species included in the Orthofinder analysis (see 'Gene family expansions and contractions' in Results) were mined for all Pfam transcription factor (TF) domains as defined in 'Table 1' of *Huang et al., 2012* and two PFAM domains [BTB (PF00651) and BACK (PF07707)] that have been implicated in transcriptional regulation (*Stogios and Privé, 2004*). Results were summarized using the *dplyr* (*Wickham et al., 2017*) and *stringr* packages (*Wickham, 2017*) within the R framework (*R Development Core Team, 2018*). Additionally, we characterized a subset of transcription factor families in greater depth [the nuclear receptor (NR), T-box, Hairy Orange, and Hox families].

### Analysis of nuclear receptors

A reciprocal BLASTp analysis (using an E-value threshold of $E^{-10}$) was performed using the *A. lycopersici*, *D. pteronyssinus*, and *T. urticae* proteomes and using the *T. urticae* nuclear receptor (NR) protein sequences as queries (*Grbić et al., 2011*) to identify putative *A. lycopersici* and *D. pteronyssinus* NRs. A tBLASTn search (using an E-value threshold of $E^{-10}$) using *T. urticae* NR protein sequences as queries (*Grbić et al., 2011*) was also performed against the *A. lycopersici* genome but only overlap with existing *A. lycopersici* NR gene models was found. LBDs of *A. lycopersici* NRs were considered present if searching with PfamScan (https://www.ebi.ac.uk/Tools/pfa/pfamscan/) or Conserved domain (CD)- search (https://www.ncbi.nlm.nih.gov/Structure/cdd/wrpsb.cgi) yielded either a PF00104 (LBD of hormone nuclear receptor) or cl11397 (The ligand binding domain of nuclear receptors, a family of ligand-activated transcription regulators) domain, respectively. Those *A. lycopersici* NRs that, in contrast to their orthologues in arthropods, were not predicted with a LBD were aligned with their orthologues in *D. melanogaster* (*Thomson et al., 2009*), *D. pteronyssinus* and *T. urticae* using the online version of MAFFT v7.380 (*Katoh et al., 2019*) with 1000 iterations and the options 'E-INS-i' and 'reorder'.

### T-box transcriptional regulators

All *A. lycopersici* T-box proteins identified in our PFAM transcription factor domain analysis were first used in BLASTp and tBLASTn searches (E-value threshold $E^{-10}$) against the *A. lycopersici* predicted proteome and genome, respectively, and no additional T-box gene models were identified. T-box proteins of *D. pteronyssinus* and *M. occidentalis* were identified in their proteomes by a BLASTp search (E-value threshold $E^{-10}$) using the conserved T-box domain amino acids 198–385 of *D. melanogaster* org-1 (FBpp0311870) as query, while those of *T. urticae* and *D. melanogaster* were derived from the PFAM analysis. *A. lycopersici* T-box proteins were aligned with those of *T. urticae*, *D.*

*pteronyssinus*, *M. occidentalis*, and *D. melanogaster* using the online version of MAFFT v7.380 (*Katoh et al., 2019*) with 1000 iterations and the options 'E-INS-i' and 'reorder' (see *Supplementary file 7*). Model selection was performed with ProtTest 3.4 (*Darriba et al., 2011*) and according to the Akaike information criterion LG+I+G+F was optimal for phylogenetic reconstruction. Next, a maximum likelihood analysis was performed using RAxML v8 HPC2-XSEDE (*Stamatakis, 2014*) on the CIPRES Science Gateway (*Miller et al., 2010*) with 1000 rapid bootstrapping replicates ('-f a -x 12345' option). The resulting tree was midpoint rooted, visualized using MEGA 6.0 (*Tamura et al., 2013*) and edited with Corel Draw software (Corel Draw, Inc).

## A. lycopersici Hairy Orange domain proteins

*A. lycopersici* and *D. pteronyssinus* orthologues of *T. urticae* Hairy Orange domain (PF07527) proteins were identified by a BLASTp search (E-value E$^{-5}$) against the *A. lycopersici* (this study) and *D. pteronyssinus* proteome (*Waldron et al., 2017*) using *T. urticae* Hairy Orange domain proteins as query. The resulting *A. lycopersici* hits were aligned with their counterparts in *D. melanogaster* (*Dearden, 2015*), *D. pteronyssinus*, and *T. urticae* using the online version of MAFFT v7.380 (*Katoh et al., 2019*) with 1000 iterations and the options 'E-INS-i' and 'reorder'.

## A. lycopersici Sox proteins

The high mobility group (HMG)-box domain (Pfam domain PF00505) of *D. melanogaster* Sox proteins (*Janssen et al., 2018*) was used as query in a BLASTp search against the *A. lycopersici*, *D. pteronyssinus*, *T. urticae*, and *M. occidentalis* proteomes. For each species, those BLASTp hits that had an E-value lower than the lowest E-value of BLASTp hits with the species orthologue of *Drosophila* capicua (a HMG-box domain protein used as outgroup in phylogenetic analysis of Sox proteins [*Janssen et al., 2018*]; aculy02g30040, g444.t1, tetur21g00740 and rna18440 in *A. lycopersici*, *D. pteronyssinus*, *T. urticae*, and *M. occidentalis*, respectively) were retained as putative Sox proteins. Almost all Acari Sox proteins contained the highly conserved RPMNAFMVW motif, characteristic of Sox proteins (*Bonatto Paese et al., 2018*); the one exception was aculy04g11170, which has a minor conservative substitution (Ala to Ser) in this motif. A tBLASTn search, using the HMG-box domain of *A. lycopersici* BLASTp hits with *D. melanogaster* Sox proteins as query, was performed to identify non-annotated *A. lycopersici* proteins; yielding one additional *A. lycopersici* Sox protein (aculy02g08510), for which a pseudogene model was created using GenomeView (*Abeel et al., 2012*). *D. pteronyssinus*, *T. urticae*, and *M. occidentalis* Sox and capicua proteins were aligned with the HMG domain of *D. melanogaster* and *P. tepidariorum* Sox proteins (*Janssen et al., 2018*) using MAFFT v7.380 (*Katoh et al., 2019*) with 1000 iterations and the options 'E-INS-i' and 'reorder'. Next, the alignment was trimmed (see *Supplementary file 7*) to contain the HMG domain only and a phylogenetic analysis of Sox HMG-box domains was performed, similar to the analysis described in *Zhong et al., 2011*. Model selection was performed with ProtTest 3.4 (*Darriba et al., 2011*) and according to the Akaike information criterion LG+I+G was optimal for phylogenetic reconstruction. A Bayesian inference was performed using MrBayes 3.2.7a (*Huelsenbeck and Ronquist, 2001*) on XSEDE on the CIPRES Science Gateway (*Miller et al., 2010*). The Monte Carlo Markov Chain search was run with four chains over 1000000 generations with trees sampled every 1000 generations. The first 250 trees were discarded as 'burn-in'. The remaining trees were used to calculate Bayesian posterior probabilities. The resulting tree was converted into a newick format using a Perl script named AfterPhylo.pl (*Zhu, 2014*), rooted with capicua proteins, visualized using MEGA 6.0 (*Tamura et al., 2013*) and edited with Corel Draw software (Corel Draw, Inc).

## Hox genes

Hox protein sequences of the oribatid mite *A. longisetosus* (*Sharma et al., 2014*), the spider mite *T. urticae* (*Grbić et al., 2011*), the deer tick *I. scapularis* (*Pace et al., 2016*) and the red flour beetle *T. castaneum* (*Pace et al., 2016*) were aligned using MAFFT v7.38 (*Katoh et al., 2019*) with 1000 iterations and the options 'L-INS-i' and 'reorder'. The 57 amino acid Homeobox domains (Pfam domain PF00046) were extracted from this alignment and used as query in a tBLASTn search (using an E-value threshold of E$^{-10}$) against the *A. lycopersici* genome to identify Hox (and by extension, also Homeobox) genes that were not automatically predicted. In one case a tBLASTn hit did show no overlap with an existing gene model and a new *A. lycopersici* Homeobox gene model (aculy01g39110) was created using GenomeView (*Abeel et al., 2012*).

To identify putative *A. lycopersici* orthologues of Hox proteins, a reciprocal BLASTp analysis (using an E-value threshold of $E^{-10}$) was performed against the *A. lycopersici* proteome (including proteins encoded by newly created gene models) using full-length *T. urticae*, *I. scapularis* and *T. castaneum* Hox protein sequences and their available proteomes (*T. urticae* version 11 August 2016, *T. castaneum* version 5.2.36 and *Ixodes scapularis* Wikel colony version 1.5). Finally, to verify the results of our reciprocal BLASTp analysis, we performed an additional BLASTp search (using an E-value threshold of $E^{-10}$) with the partial but well-studied *A. longisetosus* Hox protein sequences (*Barnett and Thomas, 2013*; *Sharma et al., 2014*; *Telford and Thomas, 1998*) as query. Using a similar approach (reciprocal BLASTp analysis with *I. scapularis* Hox proteins/*Ixodes scapularis* Wikel colony version 1.5 proteome and a BLASTp search with *A. longisetosus* Hox protein sequences), we also identified Hox protein sequences in *D. pteronyssinus*, version 2 (*Waldron et al., 2017*).

## Annotation of clock genes

Clock genes of *A. lycopersici* were identified by a tBLASTn search and reciprocal best BLASTp hit analysis (E-value threshold of $E^{-10}$) against the *A. lycopersici* genome and proteome, respectively, with *T. urticae* clock proteins (*Hoy et al., 2016*) as query.

## Prediction of the *A. lycopersici* secretome

Signal peptides of *A. lycopersici* proteins were predicted with SignalP 5.0 and using default settings (*Almagro Armenteros et al., 2019*). Transmembrane domains were predicted using the Phobius server (*Käll et al., 2007*) at http://phobius.sbc.su.se/ and protein subcellular localization was predicted using WoLF PSORT (organism type: 'Animal') at https://wolfpsort.hgc.jp/. *A. lycopersici* proteins that, according to Phobius, did not have transmembrane regions outside the 60 amino acid N-terminal region, were predicted with a signal peptide by SignalP 5.0 and were predicted to be extracellular according to Wolf PSORT, were considered as putatively secreted proteins. Putatively secreted *A. lycopersici* proteins were used as query in a BLASTp search (with E-value threshold of $E^{-10}$ and maximum target sequences set at 1) against the *T. urticae* proteome. Subsequently, *T. urticae* best BLASTp hits were mined for their presence in an LC-MS/MS analysis of *T. urticae* saliva (*Jonckheere et al., 2016*).

## miRNA identification

Mature miRNA sequences for all available arthropod species were downloaded from Release 21 of miRbase (*Kozomara and Griffiths-Jones, 2014*). miRNA sequences were aligned using STAR 2.5.2b (*Dobin et al., 2013*) to the genome of *A. lycopersici* with the following parameters '`-alignIntronMax` 0 `-alignEndsType` EndToEnd `-outFilterMismatchNmax` 2 `-outFilterMultimapNmax` 100'; this ensured that all miRNA sequences that aligned had no more than two mismatches; alignments with insertions or deletions relative to the reference were removed from further consideration, and the resulting alignment file was sorted by position and indexed using SAMtools 1.3.1 (*Li et al., 2009*). Where miRNAs from different species aligned to the same position, they were denoted as being members of the same clusters (*Supplementary file 1* — 'Table S18' Tab).

## Identification of genes in small RNA pathways

A tBLASTn search (with an E-value threshold of $E^{-5}$) using *T. castaneum* (*Prentice et al., 2015*; *Rodrigues et al., 2017*), *C. elegans* (*Hoy et al., 2016*), and *D. melanogaster* (*Iwasaki et al., 2015*) small RNA pathway-related protein sequences as query, was performed against the *A. lycopersici* genome to identify putative *A. lycopersici* small RNA pathway related genes that were not automatically predicted by the gene prediction software. As all tBLASTn hits showed overlap with existing gene models, no new gene models needed to be created. Next, a reciprocal best BLASTp hit analysis (with an E-value threshold of $E^{-5}$) was performed against the *A. lycopersici* and *T. urticae* proteome using *T. castaneum* (*Prentice et al., 2015*; *Rodrigues et al., 2017*), *C. elegans* (*Hoy et al., 2016*) and *D. melanogaster* (*Iwasaki et al., 2015*) small RNA pathway-related protein sequences and their available proteomes (*T. castaneum* version 5.2.36, *C. elegans* version WS262 and *D. melanogaster* FB2020_02 release) to identify putative small RNA pathway-related genes in *A. lycopersici* and *T. urticae*.

## Genomic HGT screen and phylogenetic validation

We performed a genomic HGT screen as previously described in *Wybouw et al., 2018*. Briefly, the *A. lycopersici* proteome was aligned with metazoan and non-metazoan proteome databases and the bitscores of the best BLASTp hits were recorded. For each protein query, the *h*-index metric was calculated by subtracting the best metazoan bitscore from the best non-metazoan bitscore. An *A. lycopersici* gene was designated as a horizontally transferred gene candidate when it exhibited a best non-metazoan bitscore $\geq$75 and an *h*-index $\geq$30. In our screen, we also performed a tBLASTn-search against the tomato russet mite scaffolds using all identified horizontally transferred *T. urticae* genes as queries. Maximum-likelihood phylogenies were subsequently constructed for all *A. lycopersici* horizontally transferred gene candidates, except for a putative UGT pseudogene that was located on scaffold 5 between coordinates 140,638 and 140,871. All complete *A. lycopersici* UGT genes were sent to the UGT Nomenclature Committee to obtain unique UGT gene names (https://prime. vetmed.wsu.edu/resources/udp-glucuronsyltransferase-homepage). For the final phylogenetic reconstruction of the pantothenate biosynthetic genes, homologues of *aculy01g38350* (ketopantoate hydroxymethyltransferase, *panB*) and *aculy04g02470* (pantoate β-alanine ligase, *panC*) were identified by BLASTn and tBLASTn searches ($E^{-10}$ cut-off) against the nonredundant nucleotide and protein NCBI databases, respectively, and were grouped based on their position in the tree of life (fungi, animals, bacteria, plants, and other). Proteins were selected per group based on manual inspection of the alignments and were combined with homologues as identified by *Wybouw et al., 2018*. In addition, we also added a panC homologue of the mealybug *Ferrisia virgata* to the final set of proteins (*Husnik and McCutcheon, 2016*). For the phylogenetic analysis of UGTs, we added UGT protein sequences from the annotated genome assembly of the house dust mite *D. pteronyssinus* (*Waldron et al., 2017*) to our UGT phylogenetic reconstruction. Applying an E-value of $E^{-10}$ as the cut-off for the alignments, 27 *D. pteronyssinus* sequences were identified by reciprocal BLASTp-searches between the *D. pteronyssinus* proteome and the 87 *T. urticae* and *A. lycopersici* UGT sequences. Protein sequences were aligned using the online version of MAFFT v7.380 (*Katoh et al., 2019*) (available at https://mafft.cbrc.jp/alignment/software/) with 1000 iterations and the options 'E-INS-i' and 'reorder' (see *Supplementary file 7*). Protein models were selected based on the Akaike Information Criterion using ProtTest 3.4 (*Darriba et al., 2011*) (panB: LG+G, panC: LG+G, and UGT: LG+G+F). Maximum likelihood analyses were performed using RAxML v8 HPC2-XSEDE (*Stamatakis, 2014*) on the CIPRES Science Gateway (*Miller et al., 2010*) with 1000 rapid bootstrap replicates ('-f a -x 12345' option). An additional maximum likelihood tree reconstruction with ultra-fast bootstrapping (1000 replicates) was performed for the pantothenate biosynthetic proteins using IQ-TREE version 1.6.12 (*Hoang et al., 2018*; *Nguyen et al., 2015*). ModelFinder identified LG+I+G4 as the best protein model based on the Bayesian Information Criterion (*Kalyaanamoorthy et al., 2017*). Constrained tree tests for alternative topologies whereby *A. lycopersici* is the sister lineage to the spider mite pantothenate biosynthetic proteins were performed using the approximately unbiased test of IQ-TREE version 1.6.12 (10,000 RELL replicates) (*Shimodaira, 2002*). The random number seed was set at 12345. Last, the physical location of the *aculy01g38350* and a*culy04g02470* genes in the *A. lycopersici* genome was examined by PCR amplification. *A. lycopersici* mites were collected by soaking infested tomato leaves overnight in 40 mL of 70% ethanol. Mites in ethanol were centrifuged at 2000 rpm for 1 min, ethanol was removed, and pelleted mites were ground using liquid nitrogen. One mL of CTAB buffer with 2% beta-mercaptoethanol and 1% proteinase K was added to the ground mites, followed by incubation in a warm water bath at 56℃. Next, samples were washed with 1 ml of choloroform:isoamyl alcohol (21:1) and DNA was precipitated with isopropanol on ice for 1 hr. Primer sequences that successfully amplified genomic regions are listed in *Supplementary file 1* — 'Table S20' Tab. PCRs were performed using the recommended protocol for Phusion High Fidelity polymerase (Thermo Scientific, The Netherlands) and 1 μL of extracted DNA (50 ng/microL) and 0.2 μM of each primer. PCR conditions for fragment 1 and 3 were 98℃ for 30 min, followed by 35 cycles of denaturation at 98℃ for 10 s, annealing at 55℃ for 30 s, and extension at 72℃ for 1 min (fragment 1) or 45 s (fragment 3) followed by a final extension step at 72℃ for 5 min. PCR conditions for fragment two were as follows: 98℃ for 30 s, 5 cycles of 98℃ for 10 s, 65℃ for 10 s, 72℃ for 60 s, five cycles of 98℃ for 10 s, 60℃ for 10 s, 72℃ for 60 s, and 20 cycles of 98℃ for 10 s, 60℃ for 10 s, 72℃ for 60 s, followed by a final extension step at 72℃ for 3 min. Resulting amplicons were Sanger sequenced by Eurofins (Leiden, The Netherlands) using PCR (with 'PCR'

suffix) and sequencing (with 'seq' suffix) primers as indicated in *Supplementary file 1* — 'Table S20' Tab.

## Acknowledgements

We thank Lin Dong and Betsie Voetdijk (University of Amsterdam, The Netherlands) for assistance with mite cultivation and PCR amplification of neighbouring genes of panB and panC, David Goldenberg (University of Utah, USA) for advice on protease annotations, Carlos Villarroel for help with the MIRA assembly, Evelien Jongepier (University of Amsterdam, The Netherlands) for assistance with data handling and data submission, Jan van Arkel (Institute for Biodiversity and Ecosystem Dynamics, The Netherlands) for providing *Figure 1a*, Ronald Ochoa (USDA-ARS) for providing an LT-SEM photograph of *A. lycopersici* (*Figure 1b*), Wendy Vanlommel (Proefcentrum Hoogstraten, Belgium) for providing *Figure 1c* and Rafael Fernández-Muñoz (Institute for Mediterranean and Subtropical Horticulture 'La Mayora', Spain) for providing *Figure 1d*. This work was supported by the Netherlands Organization for Scientific Research (STW-VIDI/13492 and STW-GAP/13550 to MRK), the USA National Science Foundation (no. 1457346 to RMC), and the European Research Council (ERC) under the European Union's Horizon 2020 research and innovation program (ERC consolidator grant 772026- POLYADAPT to TVL and 773902-SuperPests to TVL). WD and NW were supported by a Research Foundation - Flanders (FWO) postdoctoral fellowship (1274917N and 12T9818N, respectively). RG was funded in part by the National Institutes of Health genetics training grant T32GM007464.

## Additional information

### Competing interests

Merijn R Kant: Reviewing editor, *eLife*. The other authors declare that no competing interests exist.

### Funding

| Funder | Grant reference number | Author |
| --- | --- | --- |
| Netherlands Organisation for Scientific Research | STW-VIDI/13492 | Merijn R Kant |
| National Science Foundation | 1457346 | Richard M Clark |
| Horizon 2020 - Research and Innovation Framework Programme | 772026-POLYADAPT | Thomas Van Leeuwen |
| Research Foundation Flanders | 1274917N | Wannes Dermauw |
| National Institutes of Health | T32GM007464 | Robert Greenhalgh |
| Research Foundation Flanders | 12T9818N | Nicky Wybouw |
| Horizon 2020 - Research and Innovation Framework Programme | 773902-SuperPests | Thomas Van Leeuwen |
| Netherlands Organisation for Scientific Research | STW-GAP/13550 | Merijn R Kant |

The funders had no role in study design, data collection and interpretation, or the decision to submit the work for publication.

### Author contributions

Robert Greenhalgh, Data curation, Formal analysis, Investigation, Visualization, Methodology, Writing - original draft, Writing - review and editing; Wannes Dermauw, Conceptualization, Formal analysis, Investigation, Visualization, Methodology, Writing - original draft, Project administration, Writing - review and editing; Joris J Glas, Resources, Formal analysis, Investigation, Methodology; Stephane Rombauts, Data curation, Formal analysis, Methodology; Nicky Wybouw, Jainy Thomas, Ellen J

Pritham, René Feyereisen, Formal analysis, Investigation, Methodology; Juan M Alba, Resources, Investigation, Methodology; Saioa Legarrea, Resources, Investigation; Yves Van de Peer, Thomas Van Leeuwen, Conceptualization; Richard M Clark, Conceptualization, Supervision, Methodology, Writing - original draft, Project administration, Writing - review and editing; Merijn R Kant, Conceptualization, Resources, Funding acquisition, Writing - original draft, Project administration, Writing - review and editing

### Author ORCIDs

Robert Greenhalgh (iD) http://orcid.org/0000-0003-2816-3154
Wannes Dermauw (iD) https://orcid.org/0000-0003-4612-8969
Joris J Glas (iD) https://orcid.org/0000-0002-8080-4564
Stephane Rombauts (iD) https://orcid.org/0000-0002-3985-4981
Nicky Wybouw (iD) https://orcid.org/0000-0001-7874-9765
Juan M Alba (iD) http://orcid.org/0000-0003-4822-9827
Saioa Legarrea (iD) http://orcid.org/0000-0002-9127-2794
René Feyereisen (iD) http://orcid.org/0000-0002-9560-571X
Yves Van de Peer (iD) http://orcid.org/0000-0003-4327-3730
Thomas Van Leeuwen (iD) https://orcid.org/0000-0003-4651-830X
Richard M Clark (iD) https://orcid.org/0000-0002-1470-301X
Merijn R Kant (iD) https://orcid.org/0000-0003-2524-8195

### Decision letter and Author response

Decision letter https://doi.org/10.7554/eLife.56689.sa1
Author response https://doi.org/10.7554/eLife.56689.sa2

---

## Additional files

### Supplementary files

- Supplementary file 1. Supplementary Tables S1-20 as Tabs in a. xlsx file.

- Supplementary file 2. 2371 orthologous protein clusters used as input for Malin.

- Supplementary file 3. Sequence alignments and annotations of intron positions for *A. lycopersici*, *D. pteronyssinus*, *T. urticae*, *M. occidentalis*, *B. mori*, and *D. melanogaster* members of 80 orthogroups.

- Supplementary file 4. Small and large orthogroups used as input for CAFE analysis.

- Supplementary file 5. Ultrametric tree used as input for CAFE analysis.

- Supplementary file 6. Homeodomain regions of Hox protein sequences of *A. lycopersici*, *D. pteronyssinus*, *T. urticae*, *A. longisetosus*, *I. scapularis*, and *T. castaneum*.

- Supplementary file 7. Protein alignments used for phylogenetic tree construction in *Figures 2*, *4* and *6*, and the respective figure supplements.

- Transparent reporting form

### Data availability

The genomic and 454 transcriptomic datasets generated by this project are available under BioProject accessions PRJNA588358 and PRJNA588365, respectively; the Illumina transcriptome data are available under BioProject accession PRJNA588358. This Whole Genome Shotgun project has been deposited at DDBJ/ENA/GenBank under the accession WNKI00000000. The version described in this paper is version WNKI01000000. Additional datasets are hosted by the Online Resource for Community Annotation of Eukaryotes (ORCAE) at https://bioinformatics.psb.ugent.be/orcae/, where the annotation can be viewed and de novo transcriptomes (Illumina and 454) can be downloaded.

The following datasets were generated:

| Author(s) | Year | Dataset title | Dataset URL | Database and Identifier |
|---|---|---|---|---|
| Greenhalgh R, Dermauw W, Glas JJ, Rombauts S, Wybouw N, Thomas J, Alba JM, Pritham EJ, Legarrea S, Feyereisen R, Van de Peer Y, Van Leeuwen T, Clark RM, Kant MR | 2020 | Aculops lycopersici genome sequencing and assembly and Illumina transcriptome sequencing | http://www.ncbi.nlm.nih.gov/bioproject/?term=PRJNA588358 | NCBI BioProject, PRJNA588358 |
| Greenhalgh R, Dermauw W, Glas JJ, Rombauts S, Wybouw N, Thomas J, Alba JM, Pritham EJ, Legarrea S, Feyereisen R, Van de Peer Y, Van Leeuwen T, Clark RM, Kant MR | 2020 | Aculops lycopersici Transcriptome or gene expression | http://www.ncbi.nlm.nih.gov/bioproject/?term=PRJNA588365 | NCBI BioProject, PRJNA588365 |
| Greenhalgh R, Dermauw W, Glas JJ, Rombauts S, Wybouw N, Thomas J, Alba JM, Pritham EJ, Legarrea S, Feyereisen R, Van de Peer Y, Van Leeuwen T, Clark RM, Kant MR | 2020 | Aculops lycopersici, whole genome shotgun sequencing project | https://www.ncbi.nlm.nih.gov/nuccore/WNKI00000000.1/ | NCBI Nucleotide, WNKI00000000 |

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
