## [Decision Letter]

**Acceptance summary:**

The mite *Aculopslycopersici*, which has a highly reduced body plan, is an important pest of tomatoes, manipulating its host plant to facilitate feeding. This species has the smallest arthropod genome yet identified, at 32 Mb, due to remarkable patterns of DNA sequence loss. It presents a striking and unprecedented example of how a multicellular eukaryote can exist without genetic material that might be essential in most other eukaryotes.

**Decision letter after peer review:**

Thank you for submitting your article "Genome streamlining in a minute herbivore that manipulates its host plant" for consideration by *eLife*. Your article has been reviewed by two peer reviewers, and the evaluation has been overseen by a Reviewing and Senior Editor. The following individual involved in review of your submission has agreed to reveal their identity: David Heckel (Reviewer #2).

We have discussed the reviews with one another and I have drafted this decision to help you prepare a revised submission.

Summary:

This study reports on the genome of *Aculops lycopersici*, an arthropod with a highly reduced body plan. This mite species, an important pest of tomatoes, manipulates its host plant to facilitate feeding. You produced an excellent genome assembly, with almost all of the assembled DNA sequence being present in chromosomal scale scaffolds and only about 7% missing based on k-mer analyses. The major news is that *A. lycopersici* has the smallest arthropod genome yet identified, at 32.5 Mbp, due to remarkable patterns of DNA sequence loss. It presents a striking and unprecedented example of how a multicellular eukaryote can exist without genetic material that might be essential in most other eukaryotes. It is all the more remarkable in a free-living organism that has to make its living by exploiting a nutrient-deprived niche in a plant that is otherwise quite well defended against most herbivores.

The genome size reduction is associated with three notable features of this genome:

- Strong reduction in transposable elements (accounting for less than 2% of the assembled genome).

- Loss of many genes (only about ~80% of the house dust mites, and only ~55% of spider mites).

- An extreme reduction in intron number (about 10-fold, which is enormous even when accounting for the reduced number of genes).

The most notable finding is perhaps that *A. lycopersici* genes have lost most of their introns. This is very interesting and, yet, presented with almost no deeper analysis – clearly a missed opportunity, and something that could be addressed very easily, greatly enhancing the appeal of the paper.

Essential revisions:

There must have been an active mechanism for removing DNA from the genome, probably initially evolved to combat transposable elements. The lineage of higher Diptera containing *Drosophilamelanogaster* has lost many genes common to insects and indeed other Diptera, and these are few enough to be well-studied. In most cases, another *Drosophila* gene has been co-opted for the lost function (e.g. alcohol dehydrogenase, acetylcholinesterase, lost telomerase function compensated by transposable elements). Please look for orthologs of genes known to be essential in *D. melanogaster*. Are there examples of essential genes having been lost, or is the loss of gene number entirely due to contraction of gene families, i.e., most likely due to elimination of (partial) genetic redundancy?

Please add more analyses regarding the potential mechanisms of massive intron loss Some examples of potentially informative analyses are: Have introns been lost primarily through "clean" deletions? (Such a large number of precise deletions would be remarkable!) If not, do genes tend to lose or gain exonic material? Or, have introns been lost because many of these genes represent retrotransposon events? If so, the transposition event must have introduced the genes into new chromosomal and presumably regulatory regions – it should be possible to deduce this through alignments to other arachnoid genomes. For the regular reader, along with a summary of the major patterns of intron loss, illustrating examples of intron loss would be very interesting.

Additional comments:

1) There is biased loss of introns within genes, with introns toward the 5' ends of genes tending to be more likely to be retained. Retained introns are also larger, on average, than introns in other arthropods. Is this related to larger introns on averaging being under stronger evolutionary constraint and tending to be located more 5' and perhaps containing more regulatory information? (This is the case at least in *Drosophila* (Parsch J., Novozhilov S., Saminadin-Peter S. S., Wong K. M., Andolfatto P., 2010 On the utility of short intron sequences as a reference for the detection of positive and negative selection in *Drosophila*. Mol. Biol. Evol. 27: 1226-34; Haddrill P. R., Charlesworth B., Halligan D. L., Andolfatto P., 2005 Patterns of intron sequence evolution in *Drosophila* are dependent upon length and GC content. Genome Biol. 6: R67; Marais G., Nouvellet P., Keightley P. D., Charlesworth B., 2005 Intron size and exon evolution in *Drosophila*. Genetics 170: 481-485.).

2) What are your thoughts on the evolutionary timing of the pattern of gene loss? How could it have been orchestrated? Is it still ongoing?

3) What accounts for the difference between the assembly length and the k-mer estimate: is it likely to be repeats that are collapsed in the assembly (e.g., centromeric or rDNA arrays)?

4) It would be good to have a figure that illustrates the genome scaffolds, preferably with gene and TE density plotted along the scaffolds. Do these appear to be full length chromosomes? Do they have centromeres?

---

## [Author Response]

[…] The most notable finding is perhaps that A. lycopersici genes have lost most of their introns. This is very interesting and, yet, presented with almost no deeper analysis – clearly a missed opportunity, and something that could be addressed very easily, greatly enhancing the appeal of the paper.

We made a small number of other changes to fix wording issues, or to clarify a few points in the Materials and methods. For display items, and to address reviewer concerns, two panels have been added to main text Figure 2C and D, and additional figure supplements, supplementary files, and source data sets have been added to support the requested revisions. The order and naming of figures and supplementary materials has been adjusted to conform to the *eLife* format. We also checked our reported numbers throughout the Results section, and fixed a few counts for orthogroups, etc.

Essential revisions:There must have been an active mechanism for removing DNA from the genome, probably initially evolved to combat transposable elements. The lineage of higher Diptera containing Drosophila melanogaster has lost many genes common to insects and indeed other Diptera, and these are few enough to be well-studied. In most cases, another Drosophila gene has been co-opted for the lost function (e.g. alcohol dehydrogenase, acetylcholinesterase, lost telomerase function compensated by transposable elements). Please look for orthologs of genes known to be essential in *D. melanogaster*. Are there examples of essential genes having been lost, or is the loss of gene number entirely due to contraction of gene families, i.e., most likely due to elimination of (partial) genetic redundancy?

Recently Aromolaran et al., 2020, described a set of 427 genes that were considered, by several methods, to be essential for *D. melanogaster*. In our study, these essential genes coded for *D. melanogaster* members within 390 orthogroups. Forty-eight of these orthogroups did not have members within the Acariformes, the mite superorder comprising *A. lycopersici*, *D. pteronyssinus* and *T. urticae*, while for twenty-one (5.4%) orthogroups, an ortholog could *not* be identified in *A. lycopersici* while it was present in other acariform mites (Table S10 in Supplementary file 1). As outlined in our initial submission, *A. lycopersici* has the highest number of orthogroups with no members in *A. lycopersici* but at least one member in all other arthropod species. On the other hand, we also showed that *A. lycopersici* has the highest number of gene family contractions (Figure 3) and that for rapidly contracting orthogroups, with more than 10 members, 70% of these members had orthologs in their close relatives (Table S6 in Supplementary file 1). This new analysis, along with what was in our original submission (and retained in this resubmission), lend support to elimination of redundancy AND loss of some “essential genes” as assessed in other species. The fact that both are observed is not surprising, given the specialist defense suppressing lifestyle (fewer copies of detoxification may be needed) and the reduced body plan (some otherwise “essential” genes may no longer be essential). We have now included the essential gene analysis in the Results section of the manuscript (see section “Gene family contractions predominate in *A. lycopersici*”), and the results complement our discussion of gene loss as presented in the Discussion section.

Please add more analyses regarding the potential mechanisms of massive intron loss Some examples of potentially informative analyses are: Have introns been lost primarily through "clean" deletions? (Such a large number of precise deletions would be remarkable!) If not, do genes tend to lose or gain exonic material? Or, have introns been lost because many of these genes represent retrotransposon events? If so, the transposition event must have introduced the genes into new chromosomal and presumably regulatory regions – it should be possible to deduce this through alignments to other arachnoid genomes.

It is well appreciated in the field that inferring mechanisms of intron loss, in the absence of close genomes (few mutational steps, and where bp level alignments may be possible), is very challenging (for instance, see Yenerall et al., 2011, and Zhu and Niu, 2013, for examples of how mechanisms of loss could be inferred by comparative studies of close relatives). As *A. lycopersici* is highly divergent from other mites with sequenced genomes (Figure 3), our enthusiasm to undertake the types of analyses suggested by the reviewers at the time of initial submission was limited. Having said that, because of the exceptional extent of intron loss in *A. lycopersici,* we acknowledge that we should have attempted some form of the suggested analyses.

We have now done so. Briefly, we examined the impact of *A. lycopersici*-specific intron losses on predicted protein sequences encoded by a set of conserved genes in *A. lycopersici,* its closest relatives with sequenced genomes (the mites *D. pteronyssinus*, *T. urticae,* and *M. occidentalis*), and two insects with high quality genomes and annotations (that is, introns that were lost in *A. lycopersici,* but conserved in all the other species). For 97% of these losses, no insertions or deletions of amino acid residues were observed at the respective sites in encoded products. In the remaining cases (3%), small indels of a few amino acid residues in *A. lycopersici* were coincident with intron losses. Examinations of this gene set for *A. lycopersici* intron losses for which conserved introns were present in the two closest relatives, *D. pteronyssinus* and *T. urticae*, provided more events to examine, and were also overwhelmingly consistent with precise (or nearly precise) intron losses (although some of these were in regions of more ambiguous protein alignments, suggesting multiple mutational events, a confounding factor in assessing intron loss mechanisms when close genomes are not available).

As we note in a revised Discussion section, the pattern we observed is consistent with a prominent role for intron loss by recombination with reverse transcribed transcripts, although for a minority of events losses were also consistent with genomic deletions. We thank the reviewers for prompting us to do this additional analysis, which we think improves the manuscript. We have been, however, careful in our conclusions, as presented in the revised Discussion, as we feel that a number of questions about intron loss mechanisms in *A. lycopersici* cannot be definitively answered in the absence of more closely related genomes.

Finally, with respect to the following specific comment by the reviewers:

“Or, have introns been lost because many of these genes represent retrotransposon events? If so, the transposition event must have introduced the genes into new chromosomal and presumably regulatory regions – it should be possible to deduce this through alignments to other arachnoid genomes.”

Inherent to obliging this request is that there is synteny between *A. lycopersici* and the other mite genomes. Our earlier anecdotal examinations (based on specific genes) already suggested that synteny between *A. lycopersici* and the two most closely related genomes included in our analyses (*D. pteronyssinus* and *T. urticae*) was essentially nil. In response to the reviewers’ request, however, we further assessed this assumption using MCScanX (Wang et al., 2012; PMID: 22217600) and Synima (Farrer, 2017; PMID: 29162056), and found no extended regions of synteny (Author response image 1). The scant evidence that there was for micro-synteny (light grey lines in Author response image 1) was limited to a small number of tiny regions (and even there the results were tenuous, as only several genes supported each of the potential micro-synteny assignments).

**Author response image 1. respfig1:** Synteny assessment between acariform mite genomes. Synteny was assessed using Synima (PMID: 29162056). Concatenated genomes of *D. pteronyssinus*, *A. lycopersici* and *T. urticae* are shown, with possible micro-synteny with respect to *A. lycopersici,* middle, indicated by light grey connecting lines.

The divergence time between *A. lycopersici* and its closest sequenced relative (*D. pteronyssinus*) is at least ~200 million years (Xue et al., 2017; but see also Figure 3) and other cases of nearly absent local synteny between highly divergent invertebrates (hundreds of millions of years) have also been reported (for example, Ghedin et al., 2007; PMID: 17885136), so the observation of lack of synteny, even micro-synteny, is not necessarily unexpected.

We also found little to no synteny between *D. pteronyssinus* and *T. urticae* (again, for a comparison over hundreds of millions of years, Figure 3). This suggests that the processes that led to some of the striking features of *A. lycopersici*’s genome (i.e., most genes are intronless) are therefore not obligately coupled to major genomic reshuffling (again, there is no or little synteny between *D. pteronyssinus* and *T. urticae,* but both are intron “rich” compared *A. lycopersici*).

So, we have confirmed that it is therefore not possible to perform the requested genome alignments to address this specific (and logical) reviewer request.

However, another line of evidence does argue against the hypothesis of wholescale retrotransposition events (at least as an exclusive/major mechanism). While it is true that the majority of *A. lycopersici* genes are intronless, there are also genes that have lost some conserved introns but retained other conserved introns within the same gene (see the revised Results section “Features of extreme genome reduction in *A. lycopersici*”, which now explicitly notes this observation, and Table S3 in Supplementary file 1, and Supplementary file 3). This suggests intron loss “in place” in the genome (i.e., as mediated by recombination with reverse transcribed transcripts, or by genomic deletion), at least for many genes, as opposed to a massive incidence of retrotransposition (although we certainly cannot exclude some role for the latter, as raised by the reviewers). Nevertheless, we thank the reviewers for raising the retrotransposition possibility, and we now mention retrotransposition in a sentence in the revised Discussion as an additional mechanism that warrants consideration as more closely related genomes, for which there is synteny, become available in future.

For the regular reader, along with a summary of the major patterns of intron loss, illustrating examples of intron loss would be very interesting.

We agree wholeheartedly with the reviewers, and have added two panels to main text Figure 2. Figure 2C now shows an example of an arthropod conserved gene that is intronless in *A. lycopersici*, but has highly conserved introns in other species (amino acid alignments demonstrate apparent precise intron removal in all cases). Likewise, Figure 2D shows an example of a candidate imprecise *A. lycopersici* intron loss event.

Additional comments:1) There is biased loss of introns within genes, with introns toward the 5' ends of genes tending to be more likely to be retained. Retained introns are also larger, on average, than introns in other arthropods. Is this related to larger introns on averaging being under stronger evolutionary constraint and tending to be located more 5' and perhaps containing more regulatory information? (This is the case at least in *Drosophila* (Parsch J., Novozhilov S., Saminadin-Peter S. S., Wong K. M., Andolfatto P., 2010 On the utility of short intron sequences as a reference for the detection of positive and negative selection in *Drosophila*. Mol. Biol. Evol. 27: 1226-34; Haddrill P. R., Charlesworth B., Halligan D. L., Andolfatto P., 2005 Patterns of intron sequence evolution in Drosophila are dependent upon length and GC content. Genome Biol. 6: R67; Marais G., Nouvellet P., Keightley P. D., Charlesworth B., 2005 Intron size and exon evolution in Drosophila. Genetics 170: 481-485.).

This could indeed be the case (see Discussion). However, to assess potential functional sequences in introns, one would need genomes of closely related species, e.g., to infer shared but small (and often dispersed) transcription factor binding motifs under purifying selection, or otherwise levels of divergence in introns of homologous genes [as for example done by Haddrill et al. (2005) (PMID: 16086849) for the very closely related *D. melanogaster* and *D. simulans* species, for which some estimates of the divergence time have been as little as a few million years, and for which the date of the most recent common ancestor of all *Drosophila* species may be as little as ~20-40 million years (Obbard et al., 2012; PMID: 22683811)].

Unfortunately, as explained above, the genomes of such close relatives of *A. lycopersici* are currently not available. Hence, we are not able to address this particular comment using the approaches undertaken by the *Drosophila* community. However, as genomes of *A. lycopersici’s* close relatives do become available, we look forward to such analyses.

2) What are your thoughts on the evolutionary timing of the pattern of gene loss? How could it have been orchestrated? Is it still ongoing?

These are good questions. However, as *A. lycopersici* diverged at least 200 MYA from other acariform mites with sequenced genomes (see above, and Figure 3), hypotheses about the evolutionary timing and patterns of gene loss, or its orchestration, must remain speculative at this point. Our presumption, which is presented in the manuscript, is that host specialization, and suppression of plant defenses, may have allowed loss of genes in families associated with plant defense detoxification and plant host use, and that the physical miniaturization/reduction is associated with loss of other genes (including those for development of reduced or absent structures or biological processes). The question of timing is very hard to assess with existing data. We feel that definitive answers to these questions, while they should be fascinating, really do require more closely related genomes to be sequenced (at the same time, however, they highlight the importance of the *A. lycopersici* genome as a reference for a large slice of unsampled arthropod diversity and evolution).

3) What accounts for the difference between the assembly length and the k-mer estimate: is it likely to be repeats that are collapsed in the assembly (e.g., centromeric or rDNA arrays)?

For the *A. lycopersici* genome in our study, the deviation between genome assembly length and k-mer estimate was 7% ((34.81-32.51)/32.51). This deviation in size is, however, substantially smaller than the average difference of 13.3% that has been reported for three model species (*A. thaliana, C. elegans* and *D. melanogaster*; see “Supplementary Table 15” in Pflug et al., 2020; PMID: 32601059). While we could speculate about the source of the difference (to the extent that the discrepancy is even meaningful, which itself is unclear), at this point we think it best to just say we don’t know.

4) It would be good to have a figure that illustrates the genome scaffolds, preferably with gene and TE density plotted along the scaffolds. Do these appear to be full length chromosomes? Do they have centromeres?

We have now included a figure illustrating genome scaffolds with gene and TE density (Figure 2—figure supplement 2). These genome scaffolds are probably not full-length chromosomes, but approach the number of two chromosomes reported for another *Aculops* species (Helle and Wysoki, 1983). Helle and Wysoki, 1996, suggested, after observing several mitotic stages (including anaphases), that eriophyoid chromosomes are holokinetic, and probably do not possess localized centromeres (as is also true for *T. urticae*). Based on Figure 2—figure supplement 2, we found only several short regions with moderately higher repeat density and low gene density that are hallmarks of centromeres in organisms that have centromeres and pericentromeric regions. Therefore, we think our findings are most consistent with holokinetic chromosomes, as, again, purported previously for eriophyoids. In support of the reviewer’s requested analysis, the following sentence was added to the Results section:

“Across the *A. lycopersici* genome, extended regions of low genic composition and high TE density were not observed (Figure 2—figure supplement 2), consistent with the purported holocentric chromosome architecture (lack of regional centromeres) of eriophyoid mites (Helle and Wysoki, 1996).”